# Neuronal activity in sensory cortex predicts the specificity of learning in mice

Katherine C. Wood [ID] [1], Christopher F. Angeloni[1,2], Karmi Oxman[1], Claudia Clopath [ID] [3] &
Maria N. Geffen [ID] [1,2,4✉]

Learning to avoid dangerous signals while preserving normal responses to safe stimuli is essential for everyday behavior and survival. Following identical experiences, subjects exhibit fear specificity ranging from high (specializing fear to only the dangerous stimulus) to low (generalizing fear to safe stimuli), yet the neuronal basis of fear specificity remains unknown. Here, we identified the neuronal code that underlies inter-subject variability in fear specificity using longitudinal imaging of neuronal activity before and after differential fear conditioning in the auditory cortex of mice. Neuronal activity prior to, but not after learning predicted the level of specificity following fear conditioning across subjects. Stimulus representation in auditory cortex was reorganized following conditioning. However, the reorganized neuronal activity did not relate to the specificity of learning. These results present a novel neuronal code that determines individual patterns in learning.

[1] Departments of Otorhinolaryngology: Head and Neck Surgery, University of Pennsylvania, Philadelphia, PA, USA. [2] Psychology Graduate Group, University of Pennsylvania, Philadelphia, PA, USA. [3] Department of Bioengineering, Imperial College London, London, UK. [4] Departments of Neurology and Neuroscience, University of Pennsylvania, Philadelphia, PA, USA. ✉email: mgeffen@pennmedicine.upenn.edu

Learning allows our brain to adjust sensory representations based on environmental demands. Fear conditioning, in which a neutral stimulus is paired with an aversive stimulus, is a robust form of associative learning: exposure to just a few stimuli can lead to a fear response that lasts over the subject's lifetime[1,2]. However, the same fear conditioning paradigm elicits different levels of learning specificity across subjects[3–6]. In pathological cases, the generalization of the fear response to stimuli in non-threatening situations can lead to conditions such as post-traumatic stress disorder (PTSD)[7,8] and anxiety[9]. Therefore, determining the neuronal basis for learning specificity following fear conditioning is important and can lead to improved understanding of the neuropathology of these disorders. Whereas much is known about how fear is associated with the paired stimulus, the neuronal mechanisms that determine the level of specificity of fear learning remain poorly understood. Our first goal was to determine the neuronal basis for the differential fear learning specificity across subjects.

Multiple studies suggest the auditory cortex (AC) is involved in fear learning. During differential fear conditioning (DFC), inactivation of AC chemically[10], or with optogenetics[11], as well as partial suppression of inhibition in AC[12] led to decreased learning specificity using either pure tones or complex stimuli, such as FM sweeps or vocalizations[3,11–13]. These observations suggest that AC may determine the level of learning specificity. Therefore, we tested whether neuronal codes in AC prior to conditioning can predict specificity of fear learning.

The role of AC following fear conditioning is more controversial. Changes in stimulus representation in AC following association learning have been proposed to represent multiple different features of the fear response[1,13–18]. However, inactivation of the auditory cortex did not affect fear memory retrieval of pure tones[3,11]. If AC were involved in fear memory retrieval, we would expect the changes in sound representation to reflect the level of learning specificity across subjects. Therefore, our second goal was to test the role of changes in auditory cortex in shaping fear learning specificity across subjects.

To address these goals, we imaged the activity of neuronal ensembles in layers 2 and 3 of AC over weeks, before and after differential fear conditioning with pure tones. First, we established the neuronal basis for differential learning specificity across subjects by finding that neuronal activity in AC prior to fear conditioning predicted the level of learning specificity. Second, we found that the changes in stimulus representation in AC following fear conditioning were not correlated with the level of learning specificity across subjects, suggesting that the role of AC in fear learning is restricted to the consolidation period and changes in AC do not represent fear memory. These findings refine our understanding of the neuronal code for variability in fear learning across subjects and reconcile seemingly conflicting previous results on the function of the auditory cortex in fear learning.

## Results

**Learning specificity varies amongst conditioned mice**. To establish the relationship between sound-evoked activity in AC and differential fear conditioning, we recorded simultaneous neuronal activity from hundreds of neurons in AC. We tracked the same neurons before and after DFC, using two-photon imaging of a fluorescent calcium probe (GCaMP6[19], Supplementary Fig. 1, 2). Longitudinal imaging of neuronal activity in large ensembles of neurons in layers 2 and 3 of AC before and after conditioning (Fig. 1a) allowed us to compare the representation of the CS stimuli before and after learning.

We conditioned mice by exposure to 10 repeats of an alternating sequence of two tones, one of which co-terminated with a foot-shock (CS+, 15 kHz), and one which did not (CS−, 11.4 kHz). *Pseudo*-conditioned mice were presented with the same stimuli, but the foot-shock occurred during periods of silence between the stimuli (Fig. 1b). Following conditioning, we measured fear-memory retrieval by presenting the same auditory stimuli to the mice in a different context and measuring the percentage of time the mice froze during stimulus presentation and at baseline (Fig. 1c). Memory retrieval was tested after each imaging session. To test whether levels of freezing changed over retrieval sessions we fit a linear mixed-effects model to predict how freezing was affected by the retrieval session time and stimulus type. We found there was no effect of retrieval session on freezing (Fig. 1d, Supplementary Table 1, $t_{(164)} = 0.90$, $p = 0.372$) and no difference in the effect between retrieval session and stimulus type ($t_{(164)} = -1.21$, $p = 0.227$). Similarly, freezing in *pseudo*-conditioned mice was consistent over the 4 retrieval sessions (Fig. 1e, Supplementary Table 1, no effect of retrieval session or interaction (retrieval session by stimulus type), $p > 0.05$). Since there was no change in freezing over time, we do not specifically consider results with respect to the second DFC session (Fig. 1a, day 12). Henceforth we refer to DFC as the first DFC session. Conditioned mice that did not freeze to CS+ or CS− differently from baseline were excluded from subsequent analysis (5/19 mice excluded, Supplementary Fig. 3a, two-way ANOVA, $p > 0.05$, see "Methods").

Learning specificity was defined as the difference between freezing to CS+ and CS− during memory retrieval sessions (see "Methods", Eq. (1))[3]. We used two pure-tone CS stimuli which have been shown to engage AC in both mouse[3,12] and human DFC[20]. The pure tones were close together in frequency space (0.40 octaves apart) in order to drive a range of learning specificities in conditioned mice that is not achievable at greater frequency separations[3]. Indeed, we observed that conditioned mice displayed a larger range of learning specificity (range: −16.9 to 55.6%) compared with *pseudo*-conditioned mice (−4.2 to 7.6%). This was reflected in a significantly larger standard deviation of learning specificity in conditioned mice ($\sigma = 20.3\%$) than in *pseudo*-conditioned mice (Fig. 1f, $\sigma = 3.3\%$, $F$-test, $F_{(13, 8)} = 36.80$, $p < 0.001$) in the first retrieval session after DFC. We also observed a significantly higher learning specificity (mean: 15.8%) in conditioned mice than *pseudo*-conditioned mice (mean: 1.0%, $t$-test, $t_{(21)} = 2.15$, $p = 0.043$) in the retrieval session after DFC. To test whether learning specificity was consistent over retrieval sessions, we fit a linear mixed-effects model to predict how learning specificity was affected by retrieval session and conditioning type. We found no effect of retrieval session on learning specificity for conditioned mice (Supplementary Fig. 3b, c, Supplementary Table 1, $t_{(88)} = 0.23$, $p = 0.817$) nor any interaction between session and conditioning type ($t_{(88)} = -0.01$, $p = 0.995$). Thus, we found that conditioned mice exhibited a range of learning specificity, with some generalizing their fear across the CS stimuli and others specializing their fear responses to CS+. On average, the learning specificity of mice was stable over the course of the experiment.

**Neuronal responses in AC pre-DFC predict specificity of fear learning**. We used two-photon imaging to record calcium activity from neurons in auditory cortex in head-fixed mice (Fig. 2a). We presented 100-ms tone pips (frequency range: 5–32 kHz, including CS+ and CS− frequencies) to obtain frequency response functions from each neuron. We hypothesized that the activity in auditory cortex would predict learning specificity across individual mice. Thus, we tested whether neuronal discrimination of CS+ and CS− in AC pre-DFC predicted learning specificity following DFC. To assess how well single neurons could

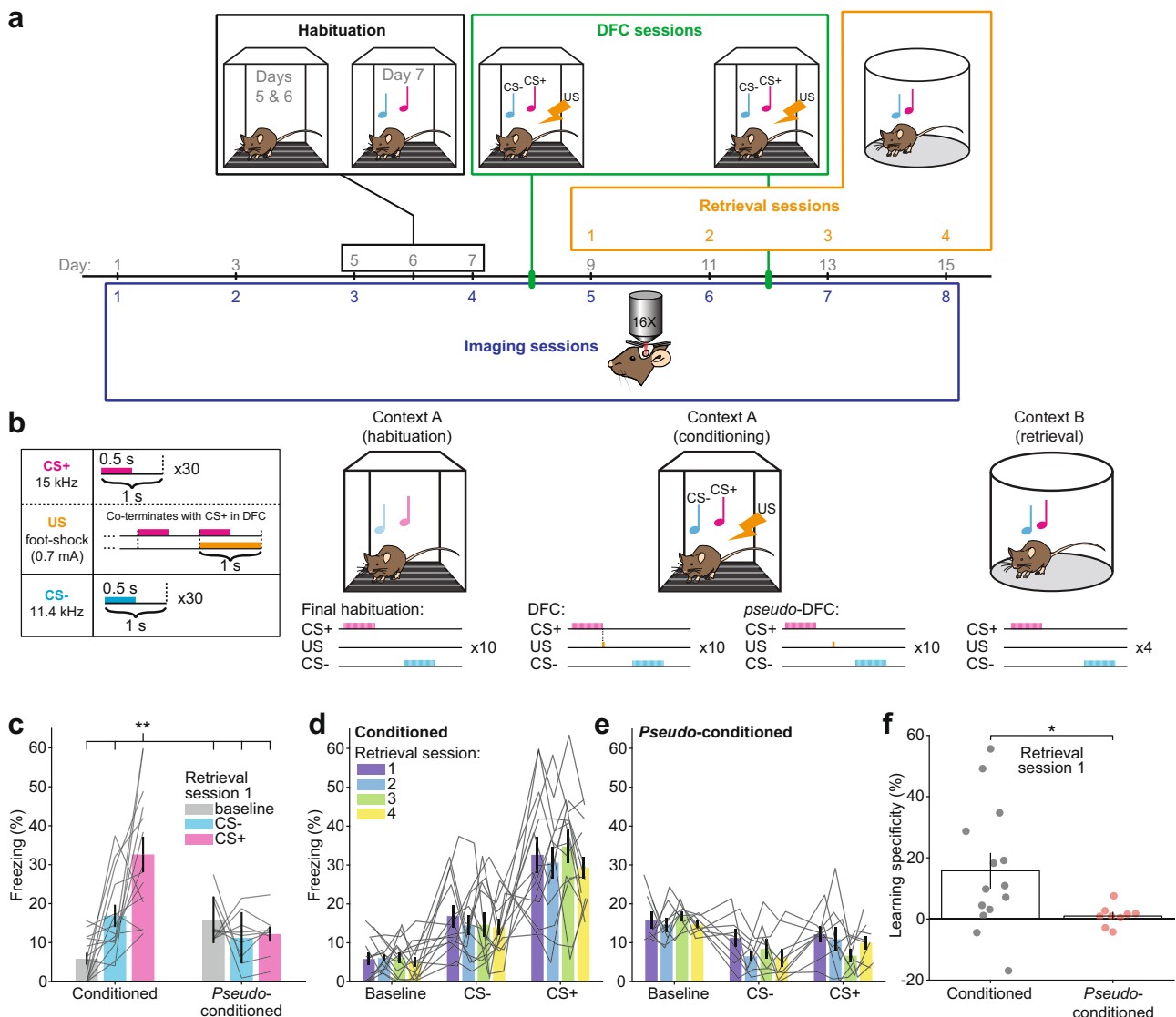

**Fig. 1 Experimental timeline and differential fear conditioning (DFC) paradigm. a** Experimental timeline: Mice were imaged for 4 sessions (48 h apart) before DFC to establish baseline responses to tone pip stimuli under the two-photon. Prior to DFC, mice were habituated to the fear conditioning chamber. Mice were subjected to DFC (19 mice) or *pseudo*-conditioning (9 mice) on Days 8 and 12. After DFC-1 (day 8), fear retrieval testing was performed after each imaging session. **b** Mice were habituated to the conditioning chamber (context A) for 3 days prior to conditioning and on the final day, the stimuli were presented without foot-shock. During conditioning, a foot-shock (1 s, 0.7 mA) was paired with the CS+ (15 kHz, 30 s pulsed at 1 Hz). The CS− (11.4 kHz, 30 s pulsed at 1 Hz) was presented alternately with the CS+ (30–180 s apart, 10 repeats) and not paired with a foot-shock. During *pseudo*-conditioning, 10 foot-shocks were presented randomly between the CS stimuli. During retrieval testing (context B), the same CS+ and CS− stimuli were presented alternately (30–180 s apart, 4 repeats). Motion of the mouse was recorded and the percentage freezing during each stimulus was measured offline. **c** Mean (±sem) percentage of time frozen during tone presentation for CS+ (pink), CS− (blue), and baseline (gray) for each mouse in retrieval session 1 (day 9). Gray lines indicate freezing for each mouse. $N = 14$ conditioned mice, $N = 9$ *pseudo*-conditioned mice. Statistics: Two-way ANOVA, Tukey–Kramer post hoc $p < 0.01$, Supplementary Table 1. **d** Mean (±sem) freezing to baseline, CS−, and CS+ for each conditioned mouse ($N = 14$) in each retrieval session. Gray lines show each mouse. **e** Same as (**d**) for *pseudo*-conditioned mice ($N = 9$). **f** Mean (±sem) learning specificity of conditioned ($N = 14$) and *pseudo*-conditioned ($N = 9$) mice for retrieval session 1. Circles show individual mice. Statistics: two-tailed, two-sample t-test, $t_{(21)} = 2.15$, $p = 0.043$. †$p < 0.1$, *$p < 0.05$, **$p < 0.01$, ***$p < 0.001$, n.s.$p > 0.05$. Source data are provided as a Source data file.

discriminate between the two conditioned tones, we computed the Z-score difference ($Z_{diff}$) of responses to CS+ and CS− for responsive neurons (see "Methods", Eq. (2)). In an example neuron (Fig. 2b), the distributions of single-trial response magnitudes to CS+ and CS− demonstrate a separation resulting in a significant $Z_{diff}$ score of 2.01. The $Z_{diff}$ score of responsive neurons was considered significant if the actual score was greater than the 95th percentile of the bootstrapped $Z_{diff}$ scores (see "Methods"). Figure 2c shows the distribution of $Z_{diff}$ scores for all responsive units from conditioned mice 24 h pre-DFC.

To test whether neuronal discrimination pre-DFC could predict subsequent learning specificity, we averaged the $Z_{diff}$ scores of neurons in each recording session of each mouse and compared it with learning specificity 24 h post-DFC. Since different numbers of neurons were recorded from each mouse, we resampled (100x with replacement) the lowest number of neurons recorded from across the mice. We found that the mean $Z_{diff}$ scores averaged across the pre-DFC imaging sessions predicted learning specificity 24 h post-DFC (Fig. 2d, Spearman's rank correlation, $r(12) = 0.81$, 95% confidence intervals (CI)

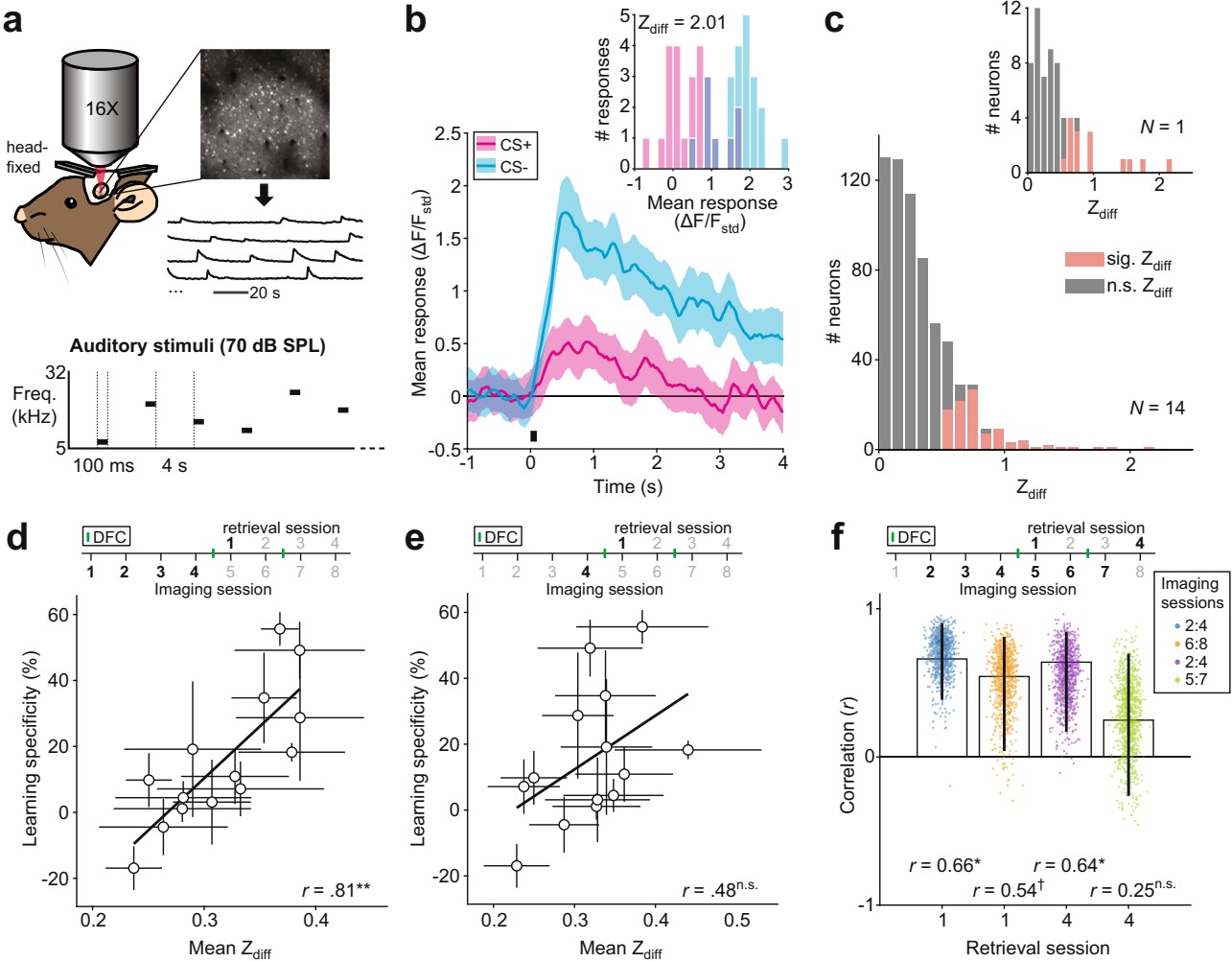

**Fig. 2 Mean neuronal discriminability pre-DFC predicts learning specificity. a** Imaging setup: Mice were head-fixed under the two-photon microscope, fluorescence of calcium indicator (GCaMP6s/m) was measured at ~30 Hz, regions of interest and mean fluorescence over time were extracted using open software[59]. Schematic showing auditory stimuli, comprised of pure-tone pips (100 ms, 70 dB SPL, 5–32 kHz) presented at 0.24 Hz. **b** Response (mean ± sem, 25 repeats) to the presentation (black bar) of CS+ (magenta) and CS− (cyan) of an example neuron. Inset shows distributions of the single-trial mean responses (mean $\Delta F/F_{std}$ across 2-s window following stimulus onset) to CS+ and CS− from the same neuron. **c** Distribution of $Z_{diff}$ scores of responsive units from conditioned mice 24 h pre-DFC. Significant scores are indicated in red, $n = 98/653$ neurons. Inset, single mouse example, $n = 15/63$ neurons. **d** Mean $Z_{diff}$ (±standard deviation [sd]) pre-DFC correlated (two-tailed Spearman's rank correlation, $r(12) = 0.81$, $p = 0.007$) with learning specificity 24 h post-DFC. Black line = linear best fit. **e** Mean $Z_{diff}$ (±sd) for each mouse 24 h pre-DFC did not correlate (two-tailed Spearman's rank correlation, $r(12) = 0.48$, $p = 0.103$) with learning specificity 24 h post-DFC. Black line = linear best fit. **f** Spearman's rank correlation (mean $r \pm 95\%$ CI) between $Z_{diff}$ score averaged across 3 imaging sessions as indicated in the legend and learning specificity from retrieval sessions 1 and 4. Dots represent individual bootstrapped $r$ ($n = 1000$). Statistics: two-tailed Spearman's rank correlation: [Imaging session 2:4, retrieval session 1] $r(12) = 0.66$, $p = 0.031$; [6:8, 1] $r(12) = 0.54$, $p = 0.066$; [2:4, 4] $r(12) = 0.64$, $p = 0.022$; [5:7, 4] $r(12) = 0.25$, $p = 0.401$. $^\dagger p < 0.1$, $^* p < 0.05$, $^{**} p < 0.01$, $^{***} p < 0.001$, $^{n.s.} p > 0.05$. Source data are provided as a Source data file.

$[0.60, 0.93]$, $p = 0.007$). Using only the imaging session preceding DFC, the mean $Z_{diff}$ did not correlate with learning specificity 24 h post-DFC (Fig. 2e, $r(12) = 0.48$, 95% CI $[0.04, 0.78]$, $p = 0.103$). However, the two correlations were not significantly different from one another (bootstrap comparison, see "Methods": $r$-difference = 0.33, 95% CI $[−0.08, 0.88]$, $p = 0.128$, $N = 14$). In summary, this suggests that the neuronal discriminability in AC of individual mice pre-DFC predicts learning specificity 24 h post-DFC.

It is possible that the $Z_{diff}$ score results from some underlying distributions of response magnitudes; for example, the magnitude of response to CS+ could be driving the prediction phenomenon. Thus, we explored whether magnitude of CS+ or CS− responses related to learning specificity. We compared the mean response magnitudes to each CS over the 4 pre-DFC imaging sessions with

learning specificity 24 h post-DFC and found that they were not correlated (Spearman's rank correlation, $p < 0.05$, Supplementary Fig. 4). This suggests that it is not merely the magnitude of responses to CS+ or CS−, but truly the discriminability of the responses that is underlying the prediction of learning specificity.

We next tested the temporal window for the prediction of learning specificity. If changes in sound-evoked responses in AC following DFC reflect memory formation or the strength of learning, as previously suggested[14,16], we would expect a stronger relationship between neuronal discrimination and learning specificity after DFC than before. To test this, we compared the correlations between mean $Z_{diff}$ across equal numbers of imaging sessions before and after DFC (3 imaging sessions preceding retrieval sessions 1, and 4) and learning specificity in retrieval sessions 1 and 4, respectively. We found that the mean $Z_{diff}$ score

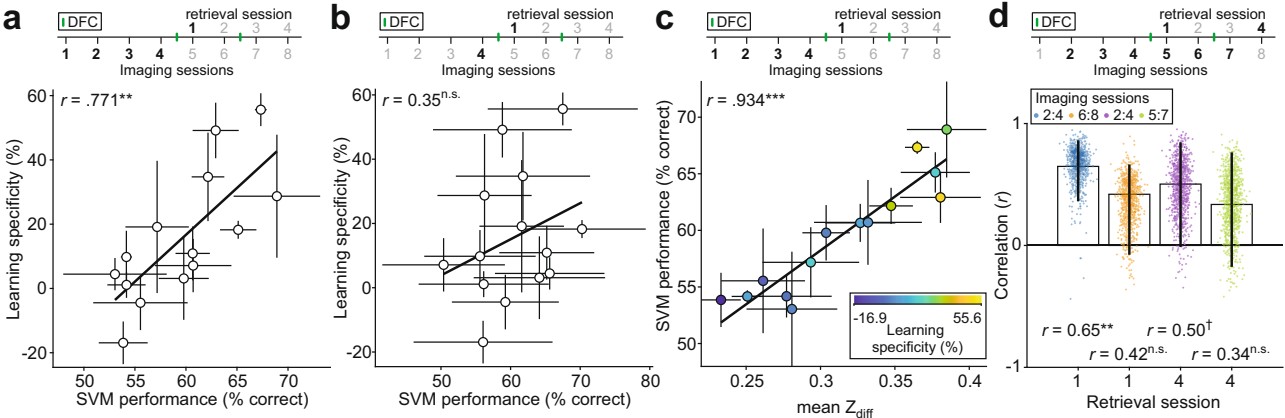

**Fig. 3 Neuronal population discrimination between CS+ and CS− pre-DFC predicts learning specificity. a** Mean (±sem) SVM performance across pre-DFC sessions predicts learning specificity 24 h post-DFC (retrieval session 1). Statistics: two-tailed Spearman's rank correlation, $r(12) = 0.77$, $p = 0.001$. **b** Mean (±sd) SVM performance 24 h pre-DFC does not predict learning specificity 24 h post-DFC. Statistics: two-tailed Spearman's rank correlation, $r(12) = 0.35$, $p = 0.247$. **c** Mean (±sem) SVM performance pre-DFC correlates with the mean (±sem) $Z_{diff}$ score pre-DFC. Fill color indicates learning specificity from retrieval session 1. Statistics: two-tailed Spearman's rank correlation, $r(12) = 0.93$, $p = <0.0001$. **d** Correlation (mean $r$ ± 95% CI) between SVM performance averaged across 3 imaging sessions preceding retrieval sessions 1 (blue), and 4 (orange). Dots represent individual bootstrapped correlation values ($n = 1000$). Statistics: two-tailed Spearman's rank correlation: [Imaging session 2:4, retrieval session 1] $r(12) = 0.65$, $p = 0.008$; [6:8, 1] $r(12) = 0.42$, $p = 0.155$; [2:4, 4) $r(12) = 0.50$, $p = 0.073$; [5:7, 4] $r(12) = 0.34$, $p = 0.252$. Black lines in **a**, **b**, and **c** show the best linear fit. $^†p < 0.1$, $^*p < 0.05$, $^{**}p < 0.01$, $^{***}p < 0.001$, $^{n.s.}p > 0.10$. Source data are provided as a Source data file.

pre-DFC predicted learning specificity from retrieval session 1 (Fig. 2f blue dots, $r(12) = 0.66$, 95% CI [0.39, 0.90], $p = 0.031$), whereas the mean $Z_{diff}$ score post-DFC did not predict learning specificity in retrieval session 4 (Fig. 2f green dots, $r(12) = 0.25$, 95% CI [−0.26, 0.70], $p = 0.401$). However, these two correlations were not significantly different (bootstrap comparison: r-difference = 0.41, 95% CI [−0.09, 0.95], $p = 0.124$). This change in prediction could result from a rearrangement of learning specificity over time or a rearrangement of $Z_{diff}$ scores over time. We reasoned that if learning specificity was rearranged then the neural discriminability pre-DFC ought not to correlate with the learning specificity in retrieval session 4 (Fig. 2f purple dots). However, we found that neural discriminability pre-DFC predicted learning specificity in retrieval session 4 ($r(12) = 0.64$, 95% CI [−0.17, 0.84], $p = 0.022$), suggesting a rearrangement of $Z_{diff}$ scores. Further supporting a rearrangement of $Z_{diff}$ scores, neuronal discriminability post-DFC did not correlate with learning specificity in retrieval session 1 (Fig. 2f orange dots, $r(12) = 0.54$, 95% CI [−0.04, 0.81], $p = 0.066$).

To verify that the results were robust to variability in frequency tuning distributions and location of the imaging window along the anterior-posterior axis (Supplementary Figs. 1 and 2) between mice, we investigated the relationship between $Z_{diff}$ and these parameters. If neuronal discriminability is affected by imaging location then we would expect a significant correlation between the location of the imaging field of view on the anterior-posterior axis and $Z_{diff}$, we did not find a significant correlation between these two measures (Supplementary Fig. 5a, Spearman's rank correlation, $p > 0.05$) nor between the percentage of neurons with significant $Z_{diff}$ and imaging location (Supplementary Fig. 5b). The best frequency distributions of neurons in the imaging window could affect the mean $Z_{diff}$ of neurons of each mouse. If so, we would expect higher $Z_{diff}$ scores and more neurons with significant $Z_{diff}$ scores for neurons tuned around the CS+ and CS−. However, we found no relationship between mean $Z_{diff}$ score and mean best frequency in the imaging window (Supplementary Fig. 5c, Spearman's rank correlation, $p > 0.05$) nor between the percentage of significant $Z_{diff}$ scores and mean best frequency (Supplementary Fig. 5d). Not surprisingly, the

percentage of significant $Z_{diff}$ scores was correlated with learning specificity (Spearman's rank correlation, $r = 0.72$, $p = 0.011$) suggesting that the best discriminating mice also had more neurons that discriminated between CS+ and CS− (Supplementary Fig. 5e). Whereas there was no relationship between mean $Z_{diff}$ and mean best frequency of neurons in the imaging window across mice, we did find that neurons with best frequency at CS+ or CS− had higher $Z_{diff}$ scores than neurons tuned to other frequencies (Supplementary Fig. 5f, Supplementary Table 1). This suggests that mice with more neurons with best frequencies at CS+ and CS− might have higher learning specificity. However, there was no relationship between the percentage of neurons in the imaging window with best frequency at CS+ and CS− across the pre-DFC imaging sessions and learning specificity (Spearman's rank correlation, $r(12) = 0.46$, 95% CI [−0.06, 0.71], $p = 0.127$).

In summary, individual neuronal discriminability in AC pre-DFC predicted learning specificity 24 h after DFC. Post-DFC, neuronal activity no longer predicted learning specificity. Therefore, the role of auditory cortex in DFC is restricted, temporally. To further investigate the relationship between neuronal and behavioral discriminability, we examined whether neuronal population discriminability could predict learning specificity.

**Population neuronal activity in AC predicts specificity of learning.** For many brain regions and tasks, activity of multiple neurons can provide more information in combination than averaged activity of individual neurons[21–23]. Using machine learning, we investigated whether populations of neurons predicted learning specificity better than the average $Z_{diff}$ scores. We trained a Support Vector Machine (SVM) to discriminate between presentation of CS+ and CS− using population responses to the two stimuli—again we resampled (100x with replacement) the lowest number of neurons recorded from across the mice. Mean SVM performance across imaging sessions prior to DFC correlated with learning specificity 24 h post-DFC (Fig. 3a, $r(12) = 0.77$, 95% CI [0.53, 0.89], $p = 0.001$). Using only the imaging session preceding DFC, SVM performance 24 h pre-DFC did not correlate with learning specificity 24 h post-DFC

(Fig. 3b, $r(12) = 0.35$, 95% CI [$-0.20$, 0.64], $p = 0.247$). However, the two correlations were not significantly different (bootstrap comparison (see "Methods"), $r$-difference $= -0.43$, 95% CI [$-0.86$, 0.00], $p = 0.056$). The $Z_{diff}$ scores and the SVM performance of the same neurons were strongly correlated (Fig. 3c, $r(12) = 0.93$, 95% CI [0.89, 0.97], $p < 0.001$), suggesting that the two different discriminability methods used similar underlying features to discriminate the stimuli. This was also reflected in the fact that the correlations between the two discriminability measures across pre-DFC imaging sessions and learning specificity were not statistically different (bootstrap comparison, $r$-difference $= 0.01$, 95% CI [$-0.09$, 0.00], $p = 0.780$). Since the SVM should give greater weight to more informative neurons, we tested whether there would be a stronger correlation between the significant $Z_{diff}$ scores and SVM performance. We found that the correlations were not significantly different (bootstrap comparison, $r$-difference $= -0.044$, 95% CI [$-0.28$, 0.14], $p = 0.562$). Thus, population responses averaged across pre-DFC imaging sessions predicted subsequent learning specificity likely through similar mechanisms to the mean $Z_{diff}$.

We next tested whether predictability of learning specificity persisted after DFC by comparing the mean SVM performance across 3 imaging sessions with retrieval sessions 1 and 4 (Fig. 3d). The mean SVM performance pre-DFC predicted learning specificity in retrieval session 1 (Fig. 3d blue dots, $r(12) = 0.65$, 95% CI [0.36, 0.86], $p = 0.008$) whereas the mean SVM performance post-DFC did not predict learning specificity in retrieval session 4 (Fig. 3d green dots, $r(12) = 0.34$, 95% CI [$-0.18$, 0.76], $p = 0.252$). However, these two correlations were not significantly different (bootstrap comparison, $r$-difference $= 0.31$, 95% CI [$-0.24$, 0.93], $p = 0.294$). Again, we tested whether this change in prediction resulted from a rearrangement of learning specificity over time or a rearrangement of neural discrimination over time. Whereas we found the same pattern of results as with $Z_{diff}$ (Fig. 2f), we did not find a significant correlation between neural discriminability pre-DFC and learning specificity in retrieval session 4 (Fig. 3d purple dots, $r(12) = 0.50$, 95% CI [$-0.01$, 0.85], $p = 0.073$). Neuronal discriminability post-DFC did not correlate with learning specificity in retrieval session 1 (Fig. 3d orange dots, $r(12) = 0.42$, 95% CI [$-0.08$, 0.66], $p = 0.155$).

Combined with similar results from the mean $Z_{diff}$ scores (Fig. 2f), these results support the interpretation that neuronal discriminability predicts learning specificity before, but not after, conditioning. This is consistent with the hypothesis that neuronal activity is reorganized following DFC and that auditory cortex can no longer modulate the freezing response following conditioning, as suggested by previous work showing that learning specificity is not dependent on auditory cortical activity after fear conditioning[3]. Since neuronal activity no longer predicted learning specificity after conditioning, we hypothesized that there would be changes in neuronal activity following conditioning. Therefore, we next investigated changes in response and neuronal discriminability following DFC.

**After DFC, neuronal discriminability between CS+ and CS− is preserved**. It has been suggested that 'fear memories' are encoded in the auditory cortex following differential fear conditioning[14,16], implying that neuronal discriminability may improve following conditioning. We found that neuronal activity following DFC no longer predicted learning specificity (Figs. 2f and 3d), suggesting AC does not support the fear response after DFC. We tested whether the neuronal discriminability of CS+ and CS− changed after DFC by comparing the mean $Z_{diff}$ across pre- and post-DFC sessions (Fig. 4a). We found no change in $Z_{diff}$ from pre- to post-

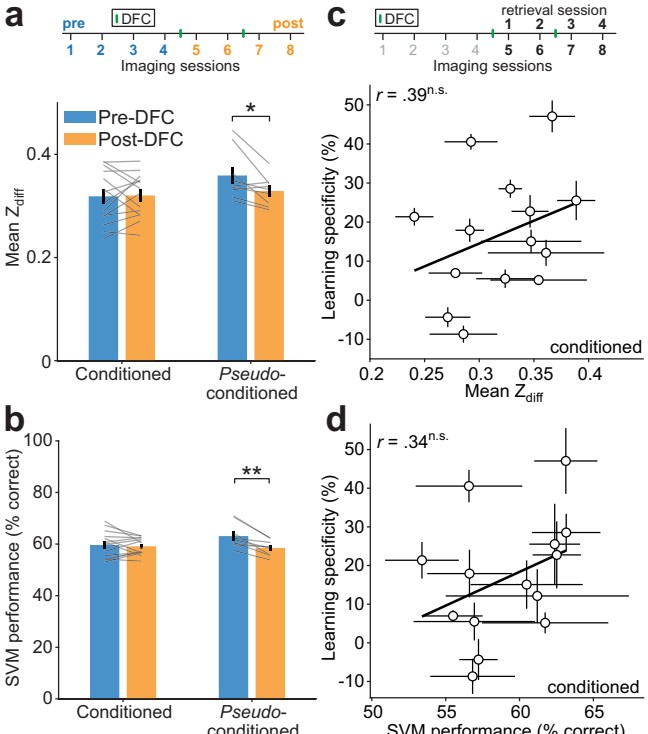

**Fig. 4 Changes in neuronal discrimination post-DFC. a** Comparison of mean ± sem $Z_{diff}$ between the pre- (sessions 1–4, blue) and post-DFC sessions (5–8, orange) in conditioned ($N = 14$) and *pseudo*-conditioned ($N = 9$) mice. Statistics: Two-way rm-ANOVA, Tukey–Kramer post hoc, $p = 0.028$, Supplementary Table 1. **b** Same as (**a**) but for comparison of mean ± sem SVM performance between the pre- and post-DFC. Statistics: Two-way rm-ANOVA, Tukey–Kramer post hoc, $p = 0.001$, Supplementary Table 1. **c** Relationship between mean (±sem) $Z_{diff}$ across the post-DFC sessions (sessions 5–8) and mean learning specificity across all retrieval sessions. Statistics: two-tailed Spearman's rank correlation, $r(12) = 0.39$, $p = 0.175$. Black line shows best linear fit. **d** Same as (**c**) but for mean (±sem) SVM performance across the post-DFC sessions and mean learning specificity. Statistics: two-tailed Spearman's rank correlation, $r(12) = 0.34$, $p = 0.264$. $^{†}p < 0.1$, $^{*}p < 0.05$, $^{**}p < 0.01$, $^{***}p < 0.001$, $^{n.s.}p > 0.10$. Source data are provided as a Source data file.

DFC in conditioned mice (Supplementary Table 1, rm-ANOVA Tukey–Kramer post hoc comparison, $p = 0.740$), whereas there was a significant decrease in *pseudo*-conditioned mice (Tukey–Kramer post hoc comparison, $p = 0.028$). Results were similar at a neuronal population level; mean SVM performance in conditioned mice did not change across pre- and post-DFC sessions (Fig. 4b, Supplementary Table 1, rm-ANOVA Tukey–Kramer post hoc comparison, $p = 0.573$), whereas there was a significant decrease in *pseudo*-conditioned mice (Tukey–Kramer post hoc comparison, $p = 0.001$). Combined, we found that following DFC or *pseudo*-conditioning, neuronal discrimination between the CS+ and CS− was maintained in conditioned mice, while it decreased in *pseudo*-conditioned mice. These results suggest that changes in AC do not improve neural discriminability. Rather, plasticity in AC in conditioned mice appeared to counteract previously reported habituation in neuronal responses to repeated stimuli[17,24].

To further investigate how neuronal discrimination changed over time, we tested the neuronal discrimination performance of the SVM using cells tracked across pairs of imaging sessions. We trained the SVM using one imaging session and obtained a baseline SVM performance on data held out from the training

session. For testing, SVM performance was measured using data from the same neurons as the training session, in the testing session (Supplementary Fig. 6a & b). If neuronal discriminability is maintained in conditioned mice, we would expect that there would be no change in performance between training and testing sessions. By contrast, in *pseudo*-conditioned mice, as neuronal discriminability appears to decrease, we expected to observe a decrease in performance particularly between sessions pre- and post-DFC. In conditioned mice, there was a small deficit in the testing sessions compared with training sessions, which did not change over different pairs of sessions. In contrast, in *pseudo*-conditioned mice, we observed the same deficit in testing sessions compared with training, but the deficit increased as the number of sessions between testing and training sessions increased. A linear regression of difference in performance with mouse group (m) and number of sessions between training and testing pairs (s) as predictors indicated that the slope of the relationship was significantly different between conditioned and *pseudo*-conditioned mice (Supplementary Table 1, m*s, $p = 0.020$). Similarly, we observed a decrease in $Z_{diff}$ as the number of sessions between pairs increased in *pseudo*-conditioned mice, but not in conditioned mice (Supplementary Fig. 6c & d, Supplementary Table 1, Linear regression, m*s, $p = 0.004$). Neuronal representations are stabilized over time with behavioral relevance and drift without[24,25]. To assess whether representation of the CS+ and CS− was stabilized in conditioned vs. *pseudo*-conditioned mice we investigated whether there was drift in the $Z_{diff}$ of populations of neurons. If there is drift in the neuronal representation, then the similarity of $Z_{diff}$ between individual neurons over time should become progressively dissimilar. We calculated the similarity (Pearson's correlation) of $Z_{diff}$ scores of neurons tracked between pairs of imaging sessions (Supplementary Fig. 6e). We fit a linear mixed-effects model to predict how $Z_{diff}$ similarity between sessions was affected by the time between imaging sessions and whether mice were conditioned or *pseudo*-conditioned. We found there was a negative effect of number of imaging sessions between pairs of sessions on $Z_{diff}$ similarity (Supplementary Table 1, $t_{(624)} = -2.87$, $p = 0.004$), but no difference in the effect between conditioned and *pseudo*-conditioned mice ($t_{(624)} = 1.28$, $p = 0.201$). In summary, there is evidence of drift in the $Z_{diff}$ score of both groups of mice, indicating that the $Z_{diff}$ of individual cells became progressively dissimilar. In conditioned mice, the average $Z_{diff}$ was maintained, while in *pseudo*-conditioned mice it decreased.

Different levels of learning specificity across mice could potentially account for the different levels of neuronal discriminability post-DFC. We therefore tested whether there was any correlation between the neuronal discriminability (mean $Z_{diff}$ score and SVM performance) and the learning specificity post-DFC. The mean $Z_{diff}$ score (imaging sessions 5–8) did not correlate with the mean learning specificity across retrieval sessions 1–4 of conditioned mice (Fig. 4c, Spearman's rank correlation, $r(12) = 0.39$, CI [−0.25, 0.73], $p = 0.175$), nor was there a correlation between the mean SVM performance post-DFC and the mean learning specificity post-DFC (Fig. 4d, $r(12) = 0.34$, CI [−0.22, 0.79], $p = 0.264$). This suggests that neuronal discriminability post-DFC does not reflect learning specificity.

**After DFC, normalized responses at CS+ increased in conditioned mice**. It has previously been shown that after differential conditioning with pure tones, select neurons in AC amplified the difference between CS+ and CS−[16,26]. However, since we observed no change in neuronal discrimination in conditioned mice, we hypothesized that there would be no change in response

to CS+ and CS−. To test whether responses were altered by conditioning, we compared frequency response functions from the pre- and post-DFC imaging sessions of responsive neurons that were tracked from pre- to post-DFC (Supplementary Fig. 7a). On an individual neuron basis, we observed heterogeneous changes in the frequency tuning (Fig. 5a, Supplementary Table 1). However, on average, in conditioned mice, the normalized response to CS+ and frequencies between the CS+ and CS− increased, whereas the response at CS− did not change (Fig. 5b, two-way rm-ANOVA, Tukey–Kramer post hoc testing, $p < 0.05$, Supplementary Table 1). In contrast, in *pseudo*-conditioned mice, the mean normalized responses at most frequencies, including both CS frequencies, did not change (Fig. 5c, Supplementary Table 1). When comparing normalized responses at CS− and CS+ in conditioned mice and the CS stimuli combined (CSc) in *pseudo*-conditioned mice, there was a significant increase at the CS+ and no change at CS− or CSc (Supplementary Table 2, Tukey–Kramer post hoc comparison, $p < 0.001$). Although we observed an increase in normalized response to CS+, there were no significant changes in non-normalized responses to conditioned frequencies in conditioned mice (Supplementary Fig. 7b, d, & e, Supplementary Table 1). We observed decreased responses to most frequencies in *pseudo*-conditioned mice (Supplementary Fig. 7c, f, & g, Supplementary Table 1). When comparing non-normalized response changes to CS+, CS− and CSc, we found a significant decrease at CSc but not at CS+ or CS− (Supplementary Table 3, Tukey–Kramer post hoc comparison, $p < 0.001$). It is likely that the normalization of the frequency response functions has amplified a small change that is not strong enough to be present in the absolute responses.

Despite the lack of significant change in the non-normalized responses, it is possible that the increase in normalized responses at CS+ and the lack of change in response at CS− in conditioned mice could lead to improved discriminability between CS+ and CS− by increasing the difference between the responses to each stimulus. This would be consistent with the hypothesis that, following fear conditioning, reorganization of neuronal activity serves to amplify the *relative* difference in responses to CS+ and CS− thereby supporting discriminability[16,18]. However, when we compared the magnitude of changes in normalized response to CS+, CS−, and the difference between the two with learning specificity, we did not find any correlation (Supplementary Fig. 8), suggesting that the changes observed are in fact not related to storage of the fear memory[27]. We further investigated by checking for a relationship between the *change* in neuronal discrimination ($Z_{diff}$ and SVM performance) and learning specificity (Supplementary Fig. 9) finding negative correlations between the two factors. This suggests that the responses of neurons that were most predictive of learning specificity changed less than responses of neurons that were less predictive, supporting the idea that reorganization of cortical activity following DFC does not depend on the fear memory, and may be due simply to random drift[25].

Previous studies found that the best frequency of neurons shifts toward the conditioned stimulus (CS+) after DFC with pure tones[16]. We observed changes in the distributions of best frequencies following DFC (Fig. 6a, b). To quantify the relationship of these changes to DFC, we calculated the absolute distance of the best frequency of responsive neurons to the CS+ frequency. Consistent with a shift in best frequency toward the CS+, we observed a small decrease in the absolute distance of best frequency from CS+ (of mean response functions pre- and post-DFC for each neuron) in responsive neurons of conditioned mice (Fig. 6c, −0.07 octaves, two-way rm-ANOVA, Tukey–Kramer post hoc, $p < 0.001$, Supplementary Table 1) but not in *pseudo*-conditioned mice ($p = 0.934$). It is possible that neuronal

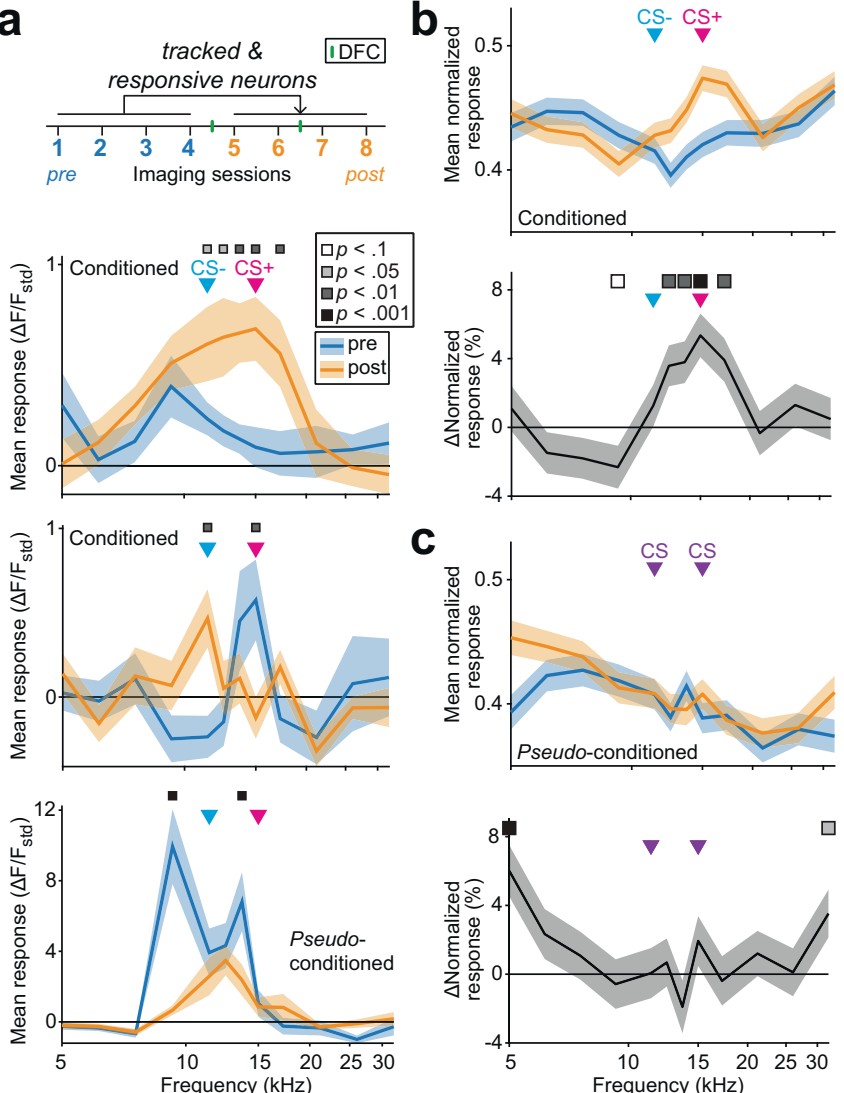

**Fig. 5 Changes in frequency representation post-DFC. a** Mean (±sem) response across repeated presentations ($N = 25$ repetitions). We tracked the responses of neurons responsive at least once pre- and post-DFC. The panels show three example frequency response functions from tracked neurons from conditioned and *pseudo*-conditioned mice pre-DFC (blue) and post-DFC (orange). Significant differences in the response functions are indicated by the squares above (two-way rm-ANOVA, Tukey–Kramer post hoc analysis, Supplementary Table 1). Arrows show the frequencies of the CS− (11.4 kHz) and CS+ (15 kHz) and CSc. Squares indicate significant changes (two-way rm-ANOVA, Tukey–Kramer post hoc analysis, Supplementary Table 1). **b** (top) Mean (±sem) normalized frequency response functions of tracked responsive neurons across all conditioned mice ($N = 14$ mice, $n = 879$ neurons). (bottom) Mean (±sem) percent change in normalized frequency response functions of the same neurons, squares indicate significant changes (two-way rm-ANOVA, Tukey–Kramer post hoc analysis, Supplementary Table 1). **c** (top) Same (**b**) for pseudo-conditioned mice ($N = 9$ mice, $n = 626$ neurons). (bottom) Percent change in normalized frequency response functions for the same cells as above, squares indicate significant changes (two-way rm-ANOVA, Tukey–Kramer post hoc analysis, Supplementary Table 1). Source data are provided as a Source data file.

discrimination between CS+ and CS− could be altered by a change in frequency tuning width[12]. As a measure of tuning width we used the sparseness of the frequency response function[28,29]: A neuron with high sparseness responds strongly to one or few frequencies tested and little to other frequencies; a neuron with a sparseness of zero would indicate an equal response to all frequencies tested. We found that sparseness decreased in both conditioned and *pseudo*-conditioned mice (Fig. 6d, two-way rm-ANOVA, $F_{(1,1503)} = 20.93$, $p < 0.001$) and that there was no difference in the magnitude of change between the two groups of mice ($F_{(1,1503)} = 0.21$, $p = 0.649$, Supplementary Table 1).

To verify that the results were robust to variability in frequency tuning between conditioned and *pseudo*-conditioned mice, we

performed the analysis on change in response, change in distance of best frequency from CS+, and change in sparseness resampling the same number of neurons from each best frequency bin (12 bins). This had the effect of normalizing the best frequency distributions pre-DFC between mice. We found that there was still an increase in response at the CS+ in conditioned mice while there were no changes at CS−, and no changes at either CS in *pseudo*-conditioned mice (Supplementary Fig. 10a). Furthermore, we found that despite the increase in response at CS+ in conditioned mice, there was no change in the absolute distance of best frequency from the CS+ while there was an increase in distance from CS+ in *pseudo*-conditioned mice (Supplementary Fig. 10b). Sparseness fell in both groups of mice (Supplementary Fig. 10c). Observing the best frequency distributions post-DFC

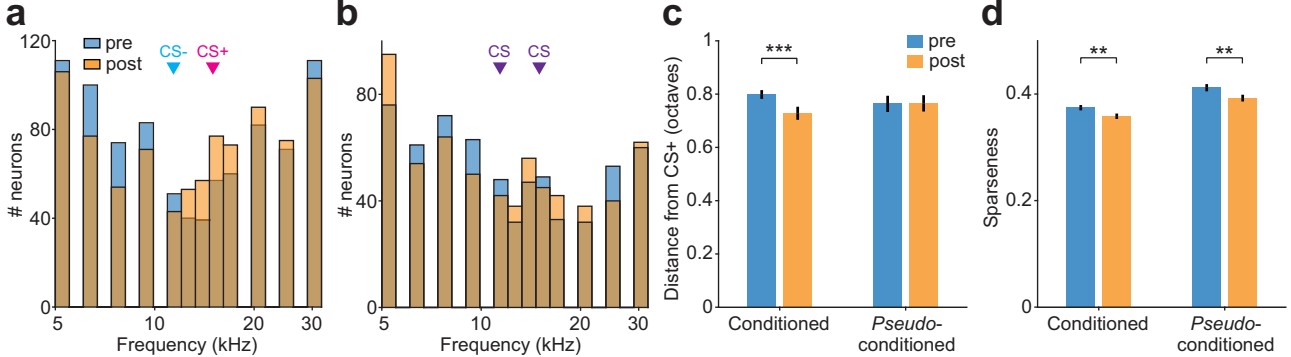

**Fig. 6 Best frequency and tuning sparseness pre- and post-conditioning. a** Distributions of best frequencies of responsive neurons pre- (blue) and post-conditioning (orange), $N = 879$. **b** Same as (**a**) for *pseudo*-conditioned mice, $N = 626$. **c** Mean (±sem) distance of best frequency from CS+ (15 kHz) of neurons from conditioned ($n = 879$, $N = 14$ mice) and *pseudo*-conditioned ($n = 626$, $N = 9$ mice) mice pre- (blue) and post-conditioning (orange). Statistics: two-way rm-ANOVA Tukey–Kramer post hoc analysis ***$p < 0.001$ (Supplementary Table 1). **d** Mean (±sem) sparseness of mean frequency response functions of neurons from conditioned ($n = 879$ neurons, $N = 14$ mice) and *pseudo*-conditioned ($n = 626$ neurons, $N = 9$ mice) mice pre- (blue) and post-conditioning (orange). Statistics: two-way rm-ANOVA Tukey–Kramer post hoc analysis **$p = 0.001$ (Supplementary Table 1). Source data are provided as a Source data file.

(Supplementary Fig. 10d), this change in best frequency in *pseudo*-conditioned mice is driven mostly by an increase in neurons with best frequency at the extremes of our measurement (5 and 32 kHz). Qualitatively, the conditioned mice showed increased numbers of neurons tuned at and above the CS+ and decreased numbers below CS+ compared with *pseudo*-conditioned mice. Thus, pairing of the CS+ with the shock led to increased number of neurons tuned to frequencies at and above the CS+ compared with an unpaired shock. Combined, whereas we find some changes in tuning consistent with classical results, these changes do not account for the individual variability in learning specificity across mice.

To investigate whether variability in the region of sampling in each mouse affected the main findings, we split the mice into two groups based on the location that the center of their imaging field of view mapped onto the anterior-posterior axis (Supplementary Fig. 2). Locations that also contained the auditory thalamus (medial geniculate body) were assigned to primary auditory cortex (A1), whereas, locations lacking MGB were assigned to anterior auditory field (AAF)[30]. We found that the changes in response at CS+ were driven by neurons in putative A1 where there was a significant increase in normalized response and not in putative AAF where there was no change in response (Supplementary Fig. 11a). The distance of best frequency from CS+ increased on average in AAF while there was no change in A1 (Supplementary Fig. 11b). However, we found no effect of imaging region on prediction of learning specificity by $Z_{diff}$ or the SVM performance pre-DFC (Supplementary Fig. 11c, d) and no effect of imaging region on change in $Z_{diff}$ from pre- to post-DFC (Supplementary Fig. 11e). Thus, there appears to be a differential effect of change in response at CS+ following conditioning for primary regions A1 and AAF, but this does not appear strongly related to the learning specificity (Supplementary Fig. 11f).

In summary, we observed heterogeneous changes in responses of individual neurons tracked from pre- to post-DFC. In conditioned animals, there was, on average, an increase in normalized response at CS+ and no change at CS−, however, the increase was not observed in non-normalized response changes. In *pseudo*-conditioned mice, we observed no changes in normalized responses at the CS stimuli. We observed a small shift in best frequency toward CS+ in conditioned mice. Sparseness of the frequency response functions decreased in both conditioned and *pseudo*-conditioned mice, indicating that frequency tuning became broader after conditioning, thus unlikely to support

improved discriminability. Combined, these results reconcile our findings with previous studies, which had effectively, by not sampling responses from the same neurons pre- and post-DFC, normalized the responses. It is plausible that previous studies observed an increase in normalized activity at CS+, which did not translate into an actual population-wide increase in discriminability.

**A learning model of the fear circuit**. We found that AC activity prior to learning predicts specificity of learning, yet the reorganized neuronal responses do not correlate with learning specificity. In order to better understand our findings in relation with previous results, we built a simple model that consisted of two frequency-tuned populations of neurons and a neuronal population that responds to the foot-shock. Our goal was to test whether this simple model could account for both the findings in this manuscript and from previous work, in particular: (1) Discriminability between CS+ and CS− in AC predicts learning specificity post-DFC (Figs. 2 and 3); (2) Suppressing inhibition in AC leads to increased generalization (decreased learning specificity) post-DFC[12]; (3) Suppressing AC post-DFC does not affect learning specificity[3,11].

In the model, we included two populations of frequency-tuned neurons (representing the medial geniculate body, MGB, and AC). MGB receives auditory inputs and projects to AC. Both populations project to basolateral amygdala (BLA). AC sends tonotopically organized feedback connections to MGB. During conditioning, the MGB neurons receive sound inputs and the neurons in the BLA are active during the foot-shock (Fig. 7a). The weights from MGB and AC to BLA are updated according to a Delta learning rule (see "Methods"), that is, they are potentiated when both are co-activated (i.e., when the foot-shock coincides with the sound stimulus). We control the level of overlap in frequency tuning between neurons in AC, which drifts over time[25], and use it to represent frequency discriminability (more overlap = less discriminability). The activity of the BLA after weight update and with auditory input only is used as a measure of freezing.

First, we first tested whether broad tuning in AC (low neuronal discriminability between CS+ and CS−) during conditioning produced more generalized freezing than sharp tuning (high neuronal discriminability). We found that increased overlap in frequency tuning in AC neurons, without changing the tuning of MGB neurons, drove more generalized freezing responses (Fig. 7b, Supplementary Fig. 12). This is due to the fact that, when AC was

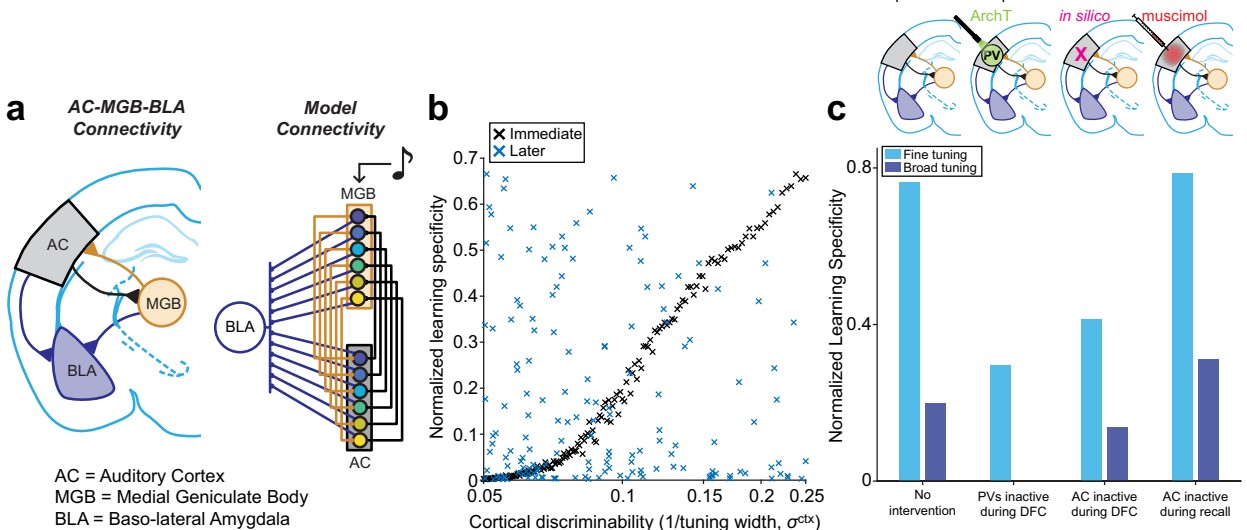

**Fig. 7 A learning model reconciles present and past findings. a** (left) Connectivity between auditory cortex (AC, gray), medial geniculate body (MGB, orange), and basolateral amygdala (BLA, blue). (Right) Model connectivity. MGB receives auditory input and provides input to AC (orange lines), and both MGB and AC provide inputs to BLA (blue lines). AC feeds back to MGB (black lines). Colored circles represent neurons tuned to different, overlapping frequency ranges. **b** Normalized learning specificity output from the model with varying levels of AC discriminability, achieved by changing the frequency tuning overlap between the neurons in the AC population, $\sigma^{ctx}$. Learning specificity was measured at two time points, immediately after DFC (black) and $10^4$ time-steps later (blue). **c** Normalized learning specificity at two AC discriminability levels; fine (light blue) and broad (dark blue) tuning. Results are shown for learning specificity with no interventions, when inhibition is reduced in AC during DFC (analog of when ArchT-transfected PV interneurons in AC are inactivated by optogenetics during DFC), when AC is inactivated during DFC (in the model), and when AC is inactivated during memory recall (analog of an injection of muscimol during memory recall; PV = parvalbumin-positive interneurons, ArchT = Archaerhodopsin-T). Source data are provided as a Source data file.

broadly tuned, CS+ tone activated AC neurons not only responding to the CS+ frequency but also to other frequencies, such as the CS−, albeit to a lesser extent. After learning, this resulted in strong AC to BLA synaptic weights that are not specific to CS+. MGB is narrowly tuned in our model, but the weights from MGB to BLA were also strengthed in a non-specific fashion because AC projects back to MGB. Therefore, CS+ also activated non-specific neurons in MGB concurrently with the foot-shock. These results support the present findings (Figs. 2 and 3). Drift in the tuning properties of the neurons in the model led to the correlation between learning specificity and tuning width decreasing over time since conditioning, consistent with our finding that neural activity post-DFC no longer predicts learning specificity (Fig. 4). Second, we examined the effects of decreasing inhibition in the AC population during conditioning (Fig. 7c, Supplementary Fig. 13). Decreasing inhibition resulted in an increased overlap in frequency responses in the AC population, which in turn led to increased generalization, supporting previous findings and providing a mechanism[12,31]. Third, we tested the effects of inactivating AC during conditioning and we found that learning specificity was reduced, consistent with the hypothesis that AC affects tone discrimination during DFC (Fig. 7c, Supplementary Fig. 14). Finally, we tested whether inactivating the auditory cortex following conditioning had an effect on freezing responses (Fig. 7c, Supplementary Fig. 15). Consistent with previous findings[3,11], we did not observe a change in fear generalization following AC inactivation. The broad or narrow tuning of AC neurons allowed for the synapses from MGB to BLA to be strengthened either narrowly or broadly during conditioning. Therefore, with suppression of AC during memory recall, the specialized versus generalized learning was preserved.

Combined, the model demonstrates that a simple anatomically consistent circuit supports multiple aspects of cortical control of fear conditioning identified here and in previous studies.

## Discussion
Our results identify the role of the auditory cortex in differential fear learning: (1) Prior to fear learning, neuronal responses in AC shape fear learning specificity (Figs. 2 and 3); (2) Following differential fear conditioning, neuronal response transformations are not correlated with fear learning specificity (Fig. 5, Supplementary Fig. 8), and therefore the auditory cortex does not encode auditory differential fear memory; (3) Neuronal activity in AC post-DFC does not correlate with freezing behavior (Fig. 4); (4) A simple model of the auditory nuclei and the basolateral amygdala could account for our results as well as a number of previous findings (Fig. 7).

Our finding that the neuronal activity prior to fear conditioning predicted specialization of fear learning provides a mechanism for the role of AC in differential fear memory acquisition[10–12,13,31]. Specifically, inactivation of inhibitory neurons in the AC during fear conditioning led to increased generalization of fear learning with pure tones[12]. Suppressing inhibitory neurons in the AC led to a decrease in Fisher information, which reflects the certainty about a stimulus in neuronal representation[31]. This change would likely result in a decrease in neuronal discriminability between the dangerous and safe tones in the AC, and therefore drive an increase in fear generalization, as demonstrated by our model (Fig. 7). Our results provide the link between optogenetic inactivation of interneurons in AC leading to increased fear generalization, and to increased frequency tuning width[12], which decreases neuronal discriminability.

By using two-photon imaging to record from the same neurons over the course of differential fear conditioning, we were able to compute changes in both absolute and relative neuronal activity of a large number of identified neurons, a feat not normally achievable with electrophysiology[16,18]. Previous work found that changes in neuronal responses to the dangerous and safe stimuli after differential fear conditioning amplified the difference

between the responses[16,18]. This change was proposed to represent fear memory[14,18,27]. We identified similar transformations in the normalized response functions of neurons that were tracked pre- to post-conditioning, finding an increased relative response to the CS+. However, these changes did not correlate with freezing behavior suggesting that the neuronal code in the AC after fear conditioning does not reflect differential fear memory. Indeed, a number of studies found that inactivating the auditory cortex after fear conditioning with pure tones does not affect fear memory retrieval[3,11] (but see ref. [13]). Combined, our results restrict the role of auditory cortex in fear conditioning to pure tone differential fear memory acquisition, but not retrieval.

If the increase in normalized response at CS+ is not related to fear memory, then why is there an increase in response? It could be reflective of increased attention caused by presentation of the CS+ and that the discrimination of the CS stimuli is unaffected by this effect[32]. Furthermore, changes in frequency map organization do not necessarily relate to changes in behavioral frequency discrimination of pure tones[33,34], thus over-representation of the CS+ could be induced by learning but not necessary for discrimination.

To locate our findings with previous work, we implemented a simple, anatomically accurate[35,36] model with connections from auditory nuclei to the basolateral amygdala (Fig. 7). The model demonstrated that (1) neuronal activity in cortex can predict subsequent learning specificity; that (2) inactivation of PV interneurons in AC during DFC leads to increased generalization[12], and that (3) the auditory cortex is not necessary for differential fear memory retrieval[3,11] and (4) that discrimination is still possible with AC inactive during conditioning but learning specificity is reduced. The model proposes that either MGB or AC or a combination of both can induce auditory fear memory through the strengthening of connections in the amygdala. We propose that feedback from auditory cortex to the MGB contributes to discrimination of perceptually similar pure tone stimuli during DFC by controlling stimulus discrimination in the MGB. This may be a direct projection neuroanatomically[35,37,38]. Random drift accounts for the lack of correlation between neuronal tuning and learning specificity after conditioning[25]. Future studies need to explore the role of the MGB and specific projections between AC, MGB, and BLA in fear learning and memory. It is likely that such an important behavioral modification as fear has redundant pathways to obtain the same behavioral outcomes[11,39–41].

Our results relied on tracking the neuronal responses in all transfected neurons in AC without distinguishing between different neuronal subtypes. Previous studies found that a specific class of inhibitory neurons increases activity with presentation of repeated tones[17,24]. It is therefore plausible that our results include a subset of neurons that function differently during fear conditioning but which we were unable to identify due to lack of selective labeling. Furthermore, we restricted our recordings to layers 2 and 3 of the auditory cortex, and it is possible our results overlook more specific changes in the thalamo-recipient layers of the cortex[42,43]. The complexity of transformations in the cortical microcircuit and between layers with learning can be explored further[44–47].

The results of the study may be restricted to pure tone stimuli. We chose pure tone stimuli because these stimuli provide a well-defined axis (frequency) along which to vary stimulus discriminability and there is strong evidence to suggest auditory cortex modulates discrimination of pure tones[33,48,49]. Furthermore, in human subjects, AC encodes threat during DFC for pure tone stimuli[20]. Our prior work established that large frequency separation between CS+ and CS− results in uniform specificity of the fear response among subjects, whereas smaller frequency separation, such as the one used here, provides for a gradient of specificity across subjects[3,12]. Other studies have found that AC is not behaviorally relevant for discrimination between pure tones separated by large frequency distances[11,50]. However, when the frequencies were brought closer together, then manipulation of AC activity did affect behavior[50]. Therefore, it is unclear whether recent conclusions that AC is involved in processing of more complex stimuli and not pure tones are due to differences in complexity of the stimulus, or to the degree to which AC can discriminate these stimuli. Furthermore, the FM sweeps used in these studies are not necessarily more complex than pure-tones for AC processing. Indeed, neurons in the inferior colliculus, which is two synapses earlier than AC, differentiate between FM sweeps, e.g., ref. [51]. Ultimately, the relevant aspect of the present study was the ability to measure how well neuronal ensembles differentiate between two stimuli. We achieved this by bringing CS+ and CS− close together in frequency, and we found that neuronal discriminability of the stimuli differed across mice and correlated with behavioral discriminability prior to DFC. We would not expect this result were the stimuli not relevant for AC. Furthermore, inactivation of AC during conditioning in the model led to decreased learning specificity (Fig. 7). Future studies will dissect to what extent the differences in neuronal codes in AC shape differential fear learning of more complex and natural sounds and its role in other forms of learning[34,50,52,53].

Our results may be applicable to understanding anxiety disorders. An extreme example of fear generalization is realized in PTSD[54]. Here we find that the present state of each individual brain, in terms of neuronal discrimination of stimuli, is predictive of the future generalization of fear in the subject. This suggests that a way to prevent generalization of dangerous and safe sounds is to improve neuronal discrimination of potentially threatening stimuli[55–58]. Further work in this area can lead to a deeper understanding how genetic and social factors, as well early life experiences, shape the role of sensory cortex in this common and devastating disorder[7,57].

We identified a neuronal correlate for inter-individual differences in learning specificity. We found that the mammalian sensory cortex plays key role in stimulus discrimination during, but not following, differential fear conditioning. These results reconcile several previous findings and suggest that the role of sensory cortex is more complex than previously thought. Investigating the changes in the cortico- and thalamo-amygdala circuit during fear learning will pave way for new findings on the mechanisms of learning and memory.

## Methods

**Mice.** All experimental procedures were in accordance with NIH guidelines and approved by the IACUC at the University of Pennsylvania. Mice were acquired from Jackson Laboratories (20 males, 10 females; PV-Cre (5) [Stock No: 017320], CamKII-Cre mice (1) [Stock No: 005359] or Cdh-23 mice (24) [Stock No: 018399]) and were housed in a room with a reversed light cycle. Experiments were carried out during the dark period. Mice were housed individually after the cranial window implant. 19 mice (13 males, 6 females) were in the conditioning group and 11 mice (7 males, 4 females) were in the *pseudo*-conditioned control group.

The Auditory Brainstem Response (ABR) to tone pips (4–32 kHz, 10–80 dB SPL) was acquired before or at the end of the experiment, when possible, in order to confirm that mice had thresholds for the stimuli at or below the presentation level (Supplementary Fig. 16). Mice with ABRs >70 dB were excluded from the study (N = 2 *pseudo*-conditioned mice, 1 PV-Cre & 1 Cdh-23) resulting in 9 *pseudo*-conditioned mice in total.

Euthanasia procedures were consistent with the recommendations of the American Veterinary Medical Association (AVMA) Guidelines on Euthanasia.

**Surgical procedures.** Mice were implanted with cranial windows over auditory cortex. Mean age of cranial window implant: 9.6 weeks [6.3–13.0 weeks]. Briefly, mice were anaesthetized with 1.5–3% isoflurane and a 3-mm circular craniotomy was performed over the left auditory cortex (stereotaxic coordinates) using a 3-mm biopsy punch centered over the stereotaxic coordinates of A1 (70% of the distance

between bregma and lambda, 4.3 mm lateral to the midline). An adeno-associated virus (AAV) vector encoding the calcium indicator GCaMP6s or GCaMP6m (AAV1.Syn.GCaMP6s.WPRE.SV40 or AAV1.Syn.GCaMP6m.WPRE.SV40, UPENN vector core) was injected (750 nl, ~$1.89 \times 10^{-12}$ genome copies·ml$^{-1}$) at a 750 μm depth from the surface of the brain at 60 nl min$^{-1}$ for expression in layer 2/3 neurons in A1. Three injections were made at the same lateral distance but separated by 0.5 mm in the anterior-posterior direction or 5 injections were made spread across the window (0.3–0.5 mm apart). The injection needle was left in place for 10 mins after the injection was complete before retraction. Injections were made using a pump (Pump 11 Elite, Harvard Apparatus, USA) and needles were pulled (P-97 Puller, Sutter Instruments, USA) from glass pipettes (Harvard Apparatus, USA) with tip openings of 30–50 μm. After injection, a circular 3-mm diameter glass coverslip (size 0 or 1, Warner Instruments) was placed in the craniotomy and fixed in place using a mix of cyanoacrylate glue and dental cement. A custom-made stainless-steel head-plate (eMachine Shop) was fixed to the skull using C&B Metabond dental cement (Parkell). The implant was further secured using black dental cement. Mice were allowed to recover for 3 days post-surgery.

**Behavioral training and testing**. Mice underwent a minimum of 4 imaging sessions (range: 4–11) prior to differential auditory fear conditioning (DFC). DFC and subsequent fear retrieval testing took place in two different contexts (A and B, discussed below). Before and after each conditioning or retrieval, we cleaned the conditioning and testing chambers with either detergent (retrieval chamber) or 70% ethanol (conditioning chamber). We recorded a video of the mouse in the testing chamber using FreezeFrame 3 software (Coulbourn) at 3.75 Hz; the subsequent movement index (mean grayscale values of frame [$n + 1$] minus the preceding frame [$n$]) was exported and analyzed offline using MATLAB. The threshold of movement was defined as the 12.5th percentile of the values from each session. The mouse was considered to be freezing if the movement index was below the threshold; the measure of freezing was expressed as a percentage of time spent freezing during stimulus presentation and for baseline during the 30 s prior to stimulus onset.

Stimuli were generated using FreezeFrame 3 and presented at 70 dB SPL from an electrostatic speaker (ES-1, TDT) mounted above the animal. DFC took place in context A (Fig. 1). Stimuli were 30 s in duration and were either a continuous pure tone (4 mice) or pulsed pure tones (500 ms duration at 1 Hz). The CS+ (15 kHz) was paired with a foot-shock (1 s, direct current, 0.7 mA, 10 pairings, inter-trial interval: 50–200 s) delivered through the floor of context A (by precision animal shocker, Coulbourn). The foot-shock either co-terminated with the continuous tone or the onset coincided with the final tone pulse of the CS+ stimuli. The CS− (11.4 kHz) was presented after each CS+-foot-shock pairing but was not reinforced (10 presentations, inter-trial interval: 20–180 s). Fear memory retrieval sessions in context B followed each two-photon imaging session after conditioning. The CS+ and CS− were presented 4 times (30 s duration, interleaved, inter-trial interval: 30–180 s). For 4 mice, longer continuous presentations of the CS+ and CS− were presented (either 120 s, 1 mouse, or 60 s, 3 mice), for these mice, trials were divided into 4 equal durations and treated as above. In *pseudo*-conditioning, the foot-shocks were presented interleaved between the stimuli in periods of silence. Baseline freezing consisted of an equal time of silence prior to tone onset.

Conditioned mice that did not freeze either to CS+ or CS− were removed from subsequent analysis (two-way ANOVA for each mouse on freezing scores to CS+, CS− and baseline from all retrieval sessions (16 trials for each CS and 32 trials for baseline). Stimulus (CS+/CS−) and baseline (stimulus/no stimulus) were the independent variables. Learners were defined as those with significant effect of baseline or baseline*stimulus, $p < 0.05$). Five mice (4 males, 1 female, all Cdh-23 strain) were excluded from the study, leaving 14 conditioned mice (9 males, 5 females).

For each mouse the learning specificity (LS, Eq. (1)[3]) was calculated as:

$$LS = \sum_{i=1}^{N} fr_{CS^+}(i)/N - \sum_{i=1}^{N} fr_{CS^-}(i)/N \qquad (1)$$

Where $i$ is the trial index, $fr_{CS^{+/-}}(i)$ is the fraction of time spent freezing during trial $i$ in the CS+/− condition, respectively, and $N$ is the number of trials per condition.

**Calcium imaging procedure and acoustic stimuli**. All imaging sessions were carried out inside a single-walled acoustic isolation booth (Industrial Acoustics). Mice were placed in the imaging setup, and the head plate was secured to a custom base (eMachine Shop) serving to immobilize the head. Mice were gradually habituated to head-fixing over 3–5 days, 3–4 weeks after surgery and before imaging commenced. Imaging took place in mice aged 19.6 ± 2.5 weeks ±sem at the end of experiments.

We recorded changes in fluorescence of GCaMP6s/m caused by fluctuations in calcium concentration in transfected neurons of awake, head-fixed mice, using two-photon microscopy (Ultima in vivo multiphoton microscope, Bruker). We used a 16X Nikon objective with 0.8 numerical aperture (Thorlabs, N16XLWD-PF). The laser (940 nm, Chameleon Ti-Sapphire) power at the brain surface was kept below 30 mW. Recordings were made at 512 × 512 pixels and 13-bit resolution at ~30 frames per second.

Stimuli were generated at a sampling rate of 400 kHz using MATLAB (MathWorks, USA) and consisted of 100-ms long tone pips in the 5–32-kHz

frequency range presented at 60–80 dB SPL. In a single recording session, each frequency was repeated 15–30 times in a *pseudo*-random order with a 4-s inter-stimulus interval.

**Cell tracking across imaging sessions**. We imaged the activity from the same cells over 15 days in layers 2/3 of auditory cortex, using blood vessel architecture, depth from the surface, and the shape of cells to return to the same imaging site. To identify regions of interest (ROI) across imaging sessions that corresponded to the same cell, the maximum-projection fluorescence images from each day were registered by transforming the coordinates of landmarks present in both images in MATLAB (2017a) using the *fitgeotrans* function. The transformation was applied to ROIs from the second imaging session to match the first—all subsequent sessions were aligned to the first imaging session. We next calculated the distance between all the pairs of centroids (mean x–y position of each ROI) across the two sessions; ROIs from the two sessions were then automatically registered as the same cell based on the nearest centroid. We then manually checked the shape and position of the ROIs for any pairs that had duplicate matches, <80% ROI overlap, or a larger than average distance between the centroid locations (>2 standard deviations). ROIs which were not matched to any earlier ROIs were counted as new cells. This process was repeated for subsequent sessions, registering the imaging field to the first session, and comparing the ROIs to the cumulative ROIs from previous sessions. A final manual inspection of all the unique ROIs was performed after all the imaging sessions were registered. ROIs that overlapped with each other extensively were excluded from the dataset since it was unclear whether they were the same or different cells. Examples of tracked cells and aligned ROIs are shown in Supplementary Fig. 1.

**Data analysis and statistical procedures**. Publicly available toolboxes[59] running on MATLAB were used to register the two-photon images, select regions of interest (ROI), and estimate neuropil contamination, resulting in a neuropil-corrected fluorescence trace ($F$) for each neuron ($F$ = trace−(neuropil*0.7)). From this corrected trace, we calculated the mean baseline fluorescence ($F_{baseline}$) and standard deviation of the baseline ($F_{std}$) over the one second prior to tone onset, and then determined the change in fluorescence over time relative to the mean baseline fluorescence ($\Delta F = F - F_{baseline}$) for each sound presentation. We then divided $\Delta F$ by $F_{std}$, effectively calculating the z-score of the fluorescence response relative to the baseline ($\Delta F/F_{std}$) for each sound presentation.

The response to each tone was defined as the mean $\Delta F/F_{std}$ over 2 s following tone onset. Neurons were deemed sound responsive if at least one of the frequency responses was different from zero (t-test, $p < 0.05$, corrected for multiple comparisons using the Holm–Bonferroni method). The frequency response function was defined as the mean response to each tone frequency across repeats. Best frequency was defined as the frequency with the highest mean response. Sparseness ($S$, Eq. (2)[28,29]) was used to estimate the sharpness of response functions, with 1 being very sharply tuned and 0 being an equal response to each tone frequency:

$$a = \frac{\left( \left( \sum r_i \right)/N \right)^2}{\sum \left( r_i^2/N \right)}, S = \frac{1-a}{1-1/N} \qquad (2)$$

where $r_i$ is the mean response to the frequency $i$, and $N$ is the total number of frequencies tested.

The Z-scored difference between responses to CS+ and CS− ($Z_{diff}$, Eq. (3)) was calculated for each neuron using the following equation:

$$Z_{diff} = \left| \frac{\sum r_{CS^+}/N - \sum r_{CS^-}/N}{\sqrt{\left( \sigma_{r_{CS^+}} \cdot \sigma_{r_{CS^-}} \right)}} \right| \qquad (3)$$

where $r_{CS^+/CS^-}$ is the single-trial mean responses (mean $\Delta F/F_{std}$ over 2 s post-stimulus onset) to CS+ and CS−, respectively, $N$ is the number of repeats of each stimulus and $\sigma$ is the standard deviation of mean responses. The $Z_{diff}$ score was considered significant if the actual $Z_{diff}$ was larger than the 95th percentile of the distribution of $Z_{diff}$ scores calculated with shuffled the CS+/CS− response labels 250 times. For mice not presented with CS+ or CS− frequencies under the two-photon, the data were linearly interpolated to estimate responses at CS− and CS+. We used average $Z_{diff}$ across pre-DFC sessions of mice to test whether there was a difference between using GCaMP6s (6/23) and GCaMP6m (17/23). We found no difference (unpaired t-test, $t(21) = 1.04$, $p = 0.309$) between the mean $Z_{diff}$ scores of the two groups of mice and thus we have analyzed them together.

For fitting the Support Vector Machine (SVM), we used MATLAB's *fitcsvm* function with a linear kernel and 10-fold cross-validation to predict the learning specificity based on the standardized single-trial population responses (mean $\Delta F/F_{std}$ over 2 s post-stimulus onset for each neuron).

We calculated the confidence intervals of Spearman's rank correlations using a bootstrap procedure, resampling, with replacement, the data 1000 times, and computing the Pearson's correlation between the resampled data. We defined the 95% confidence limits of the correlation coefficient ($r$) as the 2.5th and 97.5th percentiles of the resulting distribution of correlation coefficients. In order to assess whether two correlations were significantly different from one another we

subtracted the bootstrapped $r$ distributions of each dataset from one another, the change in $r$ was considered significant if 95% CI of the difference-distribution did not overlap with zero.

To compare results between testing groups (conditioned/*pseudo*-conditioned) we used two-way repeated measures ANOVAs, linear regressions, and linear mixed-effects models with the relevant variables (see Supplementary Tables 1–3).

For mice that were not tested at 11.4 and 15 kHz under the two-photon microscope (4 conditioned mice) responses were interpolated from the frequency response functions pre- and post-DFC. For cells present in more than one session either pre- or post-DFC, the frequency response curves from each session were averaged and the changes in response were assessed from the mean across pre- and post-DFC sessions. For comparing the fluorescence traces of responses (Supplementary Fig. 7d–g), for the 4 mice not tested directly at CS+ and CS−, the nearest frequencies were used.

**Confirming anatomical location of recording**. Upon conclusion of the imaging sessions, we removed the windows of the mice and injected a red fluorescent marker (Red Retrobeads, CTB or AAV5.CAG.hChR2(H134R)-mCherry.W-PRE.SV40 [mCherry]) into the site of imaging as identified by blood vessel patterns. Briefly, we anaesthetized mice with 1.5–3% isoflurane and used a drill (Dremel) to remove the dental cement holding the window in place. We removed the glass window and injected the red marker into the imaging site (Red Retrobeads: 250 nl, CTB: 500 nl (0.5%), mCherry: 500 nl) using a glass pipette (tip diameter: 40–50 μm) at 60 nl min$^{-1}$. Following the injection, we covered the exposed brain with silicon (Kwik-Sil, World Precision Instruments) and then coated it with dental cement. After allowing time for retrograde transport (retrobeads and CTB: 1 week) or viral transfection and expression (mCherry: 3 weeks) mice were deeply anesthetized with a mixture of Dexmedatomidine (3 mg/kg) and Ketamine (300 mg/kg) and brains were extracted following perfusion in 0.01 M phosphate buffer pH 7.4 (PBS) and 4% paraformaldehyde (PFA). They were further fixed in PFA overnight and cryopreserved in 30% sucrose solution for 2 days before slicing. The location of imaging was confirmed through fluorescent imaging (Supplementary Fig. 2). For Retrobeads and CTB, the injection site was clear as a very bright injection site, for mCherry, expression levels were measured across the AC and the site of imaging was assumed to be the section with the strongest expression/brightest red. The identified sections were cross-referenced with the Allen Institute Mouse Brain Atlas using freely available software[60].

**Model**. We simulated cortical neuronal populations, MGB populations, and a BLA neuronal population in a rate-based description of neuronal activity. We simulated $N = 10$ MGB populations. Each MGB population receives $N = 10$ inputs $x_i^{MGB}$, $i = 1...N$. To model the fact that neighboring inputs are correlated, we generated the inputs $x_i$ assuming that they each have a similar tuning to stimuli. These stimuli were modeled as 10 time-dependent activities $s_j(t)$ (which corresponded to a sound amplitude at a given frequency, $j$). The activity of input $i$ was calculated by a sum of the stimulus channels, weighted with tuning strengths

$x^{MGB}_i(t) = \sum_j T^{MGB}_{ij} s_j(t) + x^{ctx}_j(t)$. The input tuning was Gaussian: $T^{MGB}_{ij} = \left[ e^{-\frac{(i-j)^2}{2\sigma^{MGB}}} \right]_+$ for $i$ and $j$ going from 1 to 10. $[.]_+$ means that negative values are set to zeros. The term $x^{cxt}$ corresponds to the direct cortical feedback. The parameter $\sigma^{MGB}$ regulated how broad the population response is to the sound. In the model, we assumed that MGB neuronal populations always have a small overlap in neuronal responses ($\sigma^{MGB} = 0.8$).

Similarly, we simulated $N = 10$ cortical populations as $x^{ctx}_i(t) = \sum_j T^{ctx}_{ij} x^{MGB}_j(t)$.

The input tuning was also Gaussian: $T^{ctx}_{ij} = \frac{1}{1.8} \left[ e^{-\frac{(i-j)^2}{2\sigma^{ctx}}} - I^{ctx} \right]_+$ for $i$ and $j$ from 1 to 10. $I^{ctx} = 0.9$ was a broad inhibitory term.

In the simulations, we tested for two different values of initial $\sigma^{ctx}$; one corresponding to narrow tuning with a small overlap ($\sigma^{ctx} = 3$), and one corresponding to broad tuning with a large overlap ($\sigma^{ctx} = 10$). (Note that $\sigma^{MGB} = 0.8$ was equivalent to $\sigma^{ctx} = 3$ since we did not model MGB inhibition here, $I^{MGB} = 0$). To avoid boundary effects, we had a circular boundary condition of the 10 inputs, meaning that input 1 and input 10 are neighbors. We also assumed that the tuning $\sigma^{ctx}$ would drift over time. Specifically, at every time step, we added a uniform random noise between −0.25 and 0.25 to $\sigma^{ctx}$. $\sigma^{ctx}$ was bounded between 4 and 20.

Finally, we simulated one population in the BLA. It received inputs from both cortical and MGB populations, i.e., $y = w^{MGB} x^{MGB} + w^{ctx} x^{ctx}$, where $w^{MGB}$ are the weights from MGB neurons to the BLA neurons, and $w^{ctx}$ are the weights from cortical neurons to the BLA. Normalized freezing response was computed as the activity after the fear conditioning paradigm (see below) normalized by the maximal activity (i.e., when the weights are all 1).

During the fear conditioning training to simulate a CS− tone, we set (channel number 6) $s_6 = 1$, all the other inputs to zero, and a CS+ we set (channel number 3) $s_3 = 1$, all the other inputs to zero. In addition, we paired it with a shock ($e = 1$ if there is a shock, $e = 0$ otherwise). The synaptic weights were plastic under the following rules: $\triangle w^{ctx/MGB}_i = \alpha x^{ctx/MGB}_i e$, where $\alpha = 0.1$ is the learning rate. This is

analogous to the standard Delta rule. The weights were bound between 0 and 1 and are initialized at 0.1. We simulated the fear conditioning for 10 time-steps [arbitrary time] and spontaneous dynamics with tuning $\sigma^{ctx}$ drift for another 10,000 time-steps. To simulate optogenetic inactivation of PV neurons in AC[12], which decreases inhibition in AC, we lowered inhibition in AC by setting $I^{ctx} = 0.45$ (half the 'normal' level), the maximum freezing was computed with the original inhibitory term intact ($I^{ctx} = 0.9$). To simulate pharmacological inactivation of AC during memory recall (after learning), we tested the behavior of the model with AC inactivation by setting $x^{ctx}_i = 0$ during the BLA simulation protocol.

**References**. Recent work in several fields of science has identified a bias in citation practices such that papers from women and other minority scholars are under-cited relative to the number of such papers in the field[61–69]. Here we sought to proactively consider choosing references that reflect the diversity of the field in thought, form of contribution, gender, race, ethnicity, and other factors. First, we obtained the predicted gender of the first and last author of each reference by using databases that store the probability of a first name being carried by a woman[65,70]. By this measure (and excluding self-citations to the first and last authors of our current paper), our references contain 11.52% woman(first)/woman(last), 4.86% man/woman, 28.08% woman/man, and 55.54% man/man. This method is limited in that (a) names, pronouns, and social media profiles used to construct the databases may not, in every case, be indicative of gender identity and (b) it cannot account for intersex, non-binary, or transgender people. Second, we obtained predicted racial/ethnic category of the first and last author of each reference by databases that store the probability of a first and last name being carried by an author of color[71–73]. By this measure (and excluding self-citations), our references contain 9.93% author of color (first)/author of color(last), 14.5% white author/author of color, 15.09% author of color/white author, and 60.47% white author/white author. This method is limited in that (a) names and Florida Voter Data to make the predictions may not be indicative of racial/ethnic identity, and (b) it cannot account for Indigenous and mixed-race authors, or those who may face differential biases due to the ambiguous racialization or ethnicization of their names. We look forward to future work that could help us to better understand how to support equitable practices in science.

**Reporting summary**. Further information on research design is available in the Nature Research Reporting Summary linked to this article.

## Data availability
The processed data that support the findings of this study are available in Dryad at https://doi.org/10.5061/dryad.wpzgmsbhw. The raw imaging data are available upon request. Source data are provided with this paper.

## Code availability
The model code and custom-written analysis code that support the findings of this study are available in Dryad at https://doi.org/10.5061/dryad.wpzgmsbhw.

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

## Acknowledgements

The authors thank Dr. Yale Cohen, Dr. Steve Eliades, and Dr. Jay Gottfried for helpful discussions and comments on an earlier version of the manuscript. The authors also thank other members of the Geffen laboratory for their helpful advice. This work was supported by funding from the National Institute of Health grants R01DC015527, R01DC014479, and R01NS113241 to M.N.G.

## Author contributions

Conceptualization, M.N.G. and K.C.W.; methodology, K.C.W, M.N.G., C.F.A., and C.C.; software, K.C.W.; investigation, K.C.W. and K.O; formal analysis, K.C.W. and M.N.G; writing—original draft, K.C.W., C.C, and M.N.G.; writing—review and editing, K.C.W., M.N.G., and C.C.; funding acquisition, M.N.G.; resources, M.N.G., K.O., and K.C.W.; supervision, M.N.G.

## Competing interests
