## [Peer Review File · Nature Communications]

Neuronal activity in sensory cortex predicts the specificity of learning in mice.REVIEWER COMMENTS

Reviewer #1 (Remarks to the Author):

This is an interesting study that looks at how differential fear conditioning is represented at the level of the auditory cortex. Although many of the findings are intriguing, there are several major issues (highlighted below) that temper my enthusiasm to the point where I question suitability for Nature Communications.

MAJOR ISSUES

1) One particularly important aspect which seems to be lacking is the extent to which we know where the recordings are located. This is because mean significant Z_{diff} will be directly proportional to the number of neurons that have a tuned response to either the CS+ or CS-. Given that CS+ was 15kHz and CS- was 11.4kHz, any FOV that is located in high-frequency regions of either A1 or AAF, or in secondary areas of ACTx, are going to skew and effect this result. In order to interpret this correlation, we need more information regarding the anatomical location of these neurons. Although I appreciate that the authors have included gross anatomical information in S1-2, this is not enough to provide knowledge of either (1) which primary auditory field is being recorded from, be it A1 or AAF, or (2) whether neurons on the medial/lateral extent of the FOV stray into secondary regions. It's also worth noting that the brain section shown in S1-2 likely contains AAF, and not A1, as indicated. This is due to the location of the hippocampus, and the lack of MGB in the coronal section. This may be due to both the Allen Coronal Atlas and the Paxinos & Watson Atlas, not distinguishing between primary auditory fields. To mitigate the necessity for recording additional data, might it be possible to analyze tonotopic frequency gradients to identify cortical fields?

2) Although I appreciate that, in the discussion, the authors have discussed their choice to use pure tones - this is somewhat controversial in the literature. Aside from prior work showing that auditory cortex being required for complex stimulus threat memory (Dalmay et al), additional work has shown that auditory cortex is dispensible for simple behaviors that require pure tones (Ceballo et al). Given the massive thalamo-amygdalar pathway, and its role in fear condition, this raises the question as to whether the results presented here actually represent a novel cortical code, or whether they are simply inherited from sub-cortical stations like the MGB.

3) The presentation of the ABR data (Figure S12) is troublesome and actually raises more issues than it solves. In the methods it is stated that ABRs were performed to confirm that the mice had good hearing. "Good hearing" is normally quantified by utilizing an ABR to determine auditory thresholds and these are not presented here. The methods state that broadband clicks at 70 dB SPL were used. To be explicit,

you could noise expose a mouse, damaging the cochlea, eliciting a permanent shift in auditory threshold, and you would still see an ABR at 70 dB SPL. Furthermore, it is not clear what "maximum ABR amplitude" is meant to indicate - these analyses are typically carried out on individual waves. It would be beneficial to relate specific ABR measurements to learning specificity.

Minor comments

- 1) Might there be any non-biological reasons to observe a drop in responsivity? (Surgical quality and whatnot...).
- 2) Lines 168 - 178 seems like an important finding, is there an associated figure?
- 3) Can any of the changes be due to differences between GCaMP6s and m? Which mice were injected with which?
- 4) Are there any age-related differences? The methods state that, in addition to a number of Cre lines, the Cdh-23 strain was used. If the max age (27.1 weeks) applies to a Cre line with a C57 background, might that be problematic?
- 5) Figure 3 A&B seems to have the legend text switched?
- 6) Line 503. It seems like a bit of a stretch to say "We imaged neuronal activity in layers 2/3 because those are the output layers of the cortex, and therefore the plastic changes that occur within the cortex should be evident in these layers." Layer 5&6 are typically referred to as the output layers of cortex. L2/3 do contain a subset of IT-type cells, but they do not in any way reflect the bulk of cortical output.

Reviewer #2 (Remarks to the Author):

This manuscript presents a study of neural correlates of learning specificity during fear conditioning. Mice were conditioned to associate one out of two tones with a foot shock, and the specificity of the resulting conditioning was quantified by the difference in freezing following the two tones during subsequent retrieval sessions. Neural activity in the auditory cortex was recorded using calcium imaging before and after conditioning, with a large number of neurons tracked across sessions. The discriminability of the two stimuli based on neural responses was then compared with the specificity of learning across animals.

The main result of the paper is that neural discriminability in A1 before conditioning predicted the learning specificity across animals, while neural discriminability after conditioning did not.

The collected dataset is impressive, and the conclusions of the analysis are intriguing. The manuscript is mostly clearly written (with a couple of exceptions outlined below). My main issue is that I have been struggling to fit the various analyses into a coherent mechanistic view. Additional controls might help clarify the overall picture. Unfortunately, the proposed model does not help clarify potential mechanisms underlying the main finding.

Comments:

1. The main finding is that neural discriminability before differential fear conditioning (DFC) predicts learning specificity across animals in the first retrieval session, but neural discriminability after DFC did not predict learning specificity in retrieval session 4. There are two possible explanations:

(i) the neural encoding of the two sounds is different before and after DFC; comparisons of neural discriminability across session however suggest this is not the case (Fig S5).

(ii) the pattern of learning specificity across animals changes between retrieval sessions 1 and 4.

As far as I could see the second explanation was not investigated. It would be important to show how stable is the pattern of learning specificity across animals from one retrieval session to the next, and explicitly contrast the two explanations outlines above.

Additionally, I have found the analyses in Fig S5 difficult to read. It would be useful to show for example a confusion matrix where a decoder is trained on one session and tested on another.

2. It is somewhat surprising that learning specificity is better predicted by average Z-scores than by population discriminability. The explanation provided on l.213-215 does not real hold, as the SVM should give vanishing weights to neurons with non-specific responses. Conversely, the Z-score was averaged only across significant neurons. How is this statistically justified? Would the results hold if the Z-score was averaged across all neurons?

3. The model does not address the main finding of the paper, i.e. the fact that neural discriminability predicts learning specificity before but not after DFC. The model shows that higher neural discriminability leads to higher learning specificity, which seems unsurprising. The model is then used to address two findings from other papers that do not seem directly related to the main result. Unless the model is used to test a mechanism reconciling the different findings (see point 1 above), it should be left out.

4. There seems to be a mismatch between the panels D and E in Fig 2, and the corresponding legend. The legends are likely swapped, but more importantly in D "does not correlate" should presumably be "does correlate".

5. Similar issue with panels A and B in Fig 3.

6. The paragraph on l.278-286 seems to be a repetition of the result presented in the paragraph on l. 168-178. I was confused by the logic here.

7. l.88-90: there is some issue with brackets.

Reviewer #3 (Remarks to the Author):

In this manuscript, Wood et al. explore how auditory cortical activity is correlated with differential fear learning in mice. Specifically, the authors combine a tone based Differential Fear Conditioning (DFC) protocol with longitudinal two-photon imaging in the auditory cortex of mice to ask how average neural responses prior to and after DFC correlate with individual learning specificity. They observe that average neuronal discriminability between CS+ and CS- tone frequencies prior to DFC is significantly correlated with fear learning specificity. However, such a correlation is not observed post DFC suggesting that auditory cortical activity post DFC does not encode differential fear memory. Furthermore, while there are changes in the normalized neural tunings following DFC, these changes do not correlate with fear learning specificity. The authors finally use a computational model of the auditory cortex, MGB and the basolateral amygdala to test their findings and reconcile them with previous findings in this area that reinforce the role of the auditory cortex in fear learning specificity prior to but not post DFC.

While this paper explores an interesting issue, there are some key concerns that need to be addressed before these results can be fully appreciated. Additionally, the manuscript as it stands now does not take full advantage of the longitudinal imaging strategy that can be used to address deeper questions about single neuron changes pre and post DFC. Here are my concerns below:

Major concerns:

1. The authors need to make sure that their results are not biased by neuronal sampling variability across different animals. It is important to provide basic sampling information and make sure they are comparable across mice - the number of neurons sampled per mouse, number of neurons with significant zdiff per mouse, variability in best frequency distributions of sampled neurons between mice. Additionally, the authors need to ensure that their results are not biased by the region of imaging. One way to do this would be to image from multiple fields in the same mouse and compare neuron response properties across different fields, to verify if results are robust to variability in the region of sampling.

2. It is interesting that average zdiff does not change between consecutive imaging sessions, and between pre and post DFC. The authors also show that learning specificity is stable across retrieval trials post DFC. Given this general stability both at the neural and behavioral level, it is a little confusing to me that while pre Zdiff is correlated with learning specificity, post Zdiff is not. How do the authors explain this?

3. Given that the authors have measured and tracked single neuron responses longitudinally, it would be useful to see some measures of response stability/variability for individual neurons across days (both during baseline and post DFC). How stable are zdiff measures for individual neurons across baseline days and do individual neurons maintain their zdiff following DFC or are new neurons recruited? Similarly, what is the stability of frequency tuning across days for a single neuron at baseline? Do individual neurons increase their responses to CS+ tone frequency following DFC or are new neurons that were previously not responsive to CS+ recruited?

4. In figure 5, while the authors have taken population averages of neuronal tuning profiles across mice, it is important to take into consideration other changes that could be influencing population responses. For instance, do the proportions of neurons responding to CS+ or CS- change post DFC, and is this correlated with learning specificity?

5. Similarly, what are the proportions of neurons with significant zdiffs prior to and post DFC, and are they correlated to learning specificity in figures 2 and 4.

6. In Figure 6, do changes in best tone frequency distance from CS+ for a given mouse correlate with its learning specificity? Do the changes observed depend on the baseline distribution of best frequencies for a mouse?

7. The authors discuss their rationale for choosing tone stimuli in their DFC experiments as opposed to more complex auditory stimuli. The authors can more directly address the debate concerning the role of the auditory cortex in differential fear learning of simple vs complex auditory stimuli by testing one

additional question in their final model – does suppressing the auditory cortex during DFC impact learning specificity?

Minor concerns:

1. Adding to the current supplementary data in the paper, are mean response magnitudes to CS+, CS-, or their difference (normalized or absolute) post DFC correlated with learning specificity?

2. How do the authors calculate average z_{diff} values over multiple days? Do they only include neurons that can be successfully tracked and that have significant z_{diff} responses across all sessions or do these come from different populations of neurons?

3. Legends for Figure 2 D and E seem to be switched. Same for Figure 3A and B.

We submit a revised manuscript “Neuronal activity in sensory cortex predicts the specificity of learning”. This manuscript was previously reviewed at Nature Communications as NCOMMS-21-04854T. We thank the reviewers for their thoughtful and insightful reviews of our manuscript. Whereas the reviewers recognized the importance of the findings, they also had a number of important methodological concerns. We have extensively revised the manuscript, including additional data, analysis, and discussion.

Major changes to the paper include an extensive re-analysis of the anatomical data for the location of imaging sites and its effect on the results (Figures S1, S2, S5, S11); re-analyzed the data with different statistical methods and resampling as suggested by reviewers (Figures 2, 3, 4, S10); analyzed the possible alternative explanations to the results as suggested by the reviewers (Figures S5, S6, S8, S10, S16); and extended the computational model and discussion of the potential mechanism (Figure 7, S14). We believe that we have addressed each and every one of the reviewers’ concerns, and as a result we present a stronger manuscript. We detail the specific revisions that address reviewers’ concerns below.

Reviewer #1 (Remarks to the Author):

This is an interesting study that looks at how differential fear conditioning is represented at the level of the auditory cortex. Although many of the findings are intriguing, there are several major issues (highlighted below) that temper my enthusiasm to the point where I question suitability for Nature Communications.

MAJOR ISSUES

1. One particularly important aspect which seems to be lacking is the extent to which we know where the recordings are located. This is because mean significant Z_{diff} will be directly proportional to the number of neurons that have a tuned response to either the CS+ or CS-. Given that CS+ was 15kHz and CS- was 11.4kHz, any FOV that is located in high-frequency regions of either A1 or AAF, or in secondary areas of ACtx, are going to skew and effect this result.

We have directly addressed the questions of best frequency distributions and relationship of Z_{diff} to best frequency and anterior-posterior location. Please see the response to Reviewer 3 point 1. We have also included the best frequency distributions for each mouse in Fig S1.

In order to interpret this correlation, we need more information regarding the anatomical location of these neurons. Although I appreciate that the authors have included gross anatomical information in S1-2, this is not enough to provide knowledge of either (1) which primary auditory field is being recorded from, be it A1 or AAF, or (2) whether neurons on the medial/lateral extent of the FOV stray into secondary regions. It's also worth noting that the brain section shown in S1-2 likely contains AAF, and not A1, as indicated. This is due to the location of the hippocampus, and the lack of MGB in the coronal section. This may be due to both the Allen Coronal Atlas and the Paxinos & Watson Atlas, not distinguishing between primary auditory fields. To mitigate the necessity for recording additional data, might it be possible to analyze tonotopic frequency gradients to identify cortical fields?

To better localize the imaging sites, we have conducted additional analysis of the histological and imaging data. In figure R1, the center of the imaging field of view (FOV) on the surface of the brain of conditioned (yellow) and pseudo-conditioned (blue) mice ($N = 26/28$) is indicated on the mouse brain atlas (Franklin & Paxinos, 3rd edition (2007)). The left stack of sections contains the MGB and thus it is most likely that the primary auditory area (Au1) represents A1 (Hackett et al. 2011, J. Neurosci.). The more anterior right-hand stack of sections does not contain the MGB and thus Au1 more likely represents the Anterior auditory field (AAF), especially in more ventral locations. However, even at more anterior locations, the dorsal auditory regions (AuD) probably include A1 based on the proposed map of auditory areas by Romero et al. (Romero et al. 2020, Cerebral Cortex). Romero and colleagues also note that the most ventral/lateral edge of the tonotopic auditory fields also lies more ventral than indicated by this brain atlas and thus it is plausible that the medial part of AuV represents A1 in the posterior sections. This is now included in Figure S2.

Figure R1. Using the red injection sites as markers for the location of the imaging windows, the brain sections were aligned with the mouse brain atlas (Franklin and Paxinos 2007) to obtain the anterior-posterior location of the imaging windows relative to bregma and the field of auditory cortex imaged from. The center of the imaging field of view on the surface of the brain of conditioned (yellow) and pseudo-conditioned (blue) mice ($N = 26/28$) is indicated on the mouse brain atlas adapted from. The left stack of sections contains the MGB, and the more anterior right-hand stack of sections does not contain the MGB Franklin & Paxinos, 3rd edition Mouse brain atlas.

Figure R2. Top left: image of the brain illuminated with blue light. Yellow box indicates the location of the imaging fov. Bottom left: Imaging fov (average of Z-stack measured at $10\mu\text{m}$ intervals of $\sim 0-150\mu\text{m}$ from the surface of the brain with two-photon imaging) visible in place on the brain surface. Blood vessel patterns were used to locate the imaging plane in the widefield fov. Top right: Same image from top left with thresholded responses of individual pixels to low (5 kHz, blue), medium (15kHz, green) and high (30kHz, red) frequency tone presentations. Bottom right: Proposed auditory cortex schematic from Romero et al. (2020) superimposed indicated approximate cortical fields.

For mice lacking obvious red markers for the site of the 2P-imaging FOV (2 mice), we examined the widefield imaging results and used responses to low, medium, and high frequency stimuli to identify the auditory fields of the imaging windows. Below is an example widefield result from one of these mice (Figure R2). Left top – Image of the surface blood vessels from widefield imaging with the 2P-imaging FOV superimposed (yellow). Left bottom

– same as above but with the average 2P image from ~100 μ m below the brain surface overlaid. Middle top – Grayscale image shows widefield imaging FOV. The filled shapes represent pixels responding above threshold to 5 kHz (blue), 15 kHz (green) and 30 kHz (red) tones. Middle bottom – Same FOV with the schematic map from Romero et al. (2020) superimposed (black outline, 2mm², arrows represent the direction of tonotopy from low (L) to high (H)) and the imaging FOV outlined in yellow. Right top – the anterior (A) – posterior (P) and medial (M) – lateral (L) directions. Right bottom – the auditory cortex schematic from Romero et al. (2020) with colors representing frequency tuning (blue = low, red = high).

In summary, we imaged from 28 mice with the following imaging locations: 17 A1, 1 SRAF, 10 AAF. Non-learners excluded following conditioning were 3 A1, 1 AAF, 1 SRAF. We have updated figure S1 (previously S2) with this analysis.

1) Although I appreciate that, in the discussion, the authors have discussed their choice to use pure tones - this is somewhat controversial in the literature. Aside from prior work showing that auditory cortex being required for complex stimulus threat memory (Dalmay et al), additional work has shown that auditory cortex is dispensable for simple behaviors that require pure tones (Ceballo et al). Given the massive thalamo-amygdalar pathway, and its role in fear condition, this raises the question as to whether the results presented here actually represent a novel cortical code, or whether they are simply inherited from sub-cortical stations like the MGB.

There is strong prior evidence to suggest that auditory cortex modulates discrimination of pure tone (Aizenberg et al. 2015; Dykstra et al. 2012; Talwar and Gerstein 2001; Tramo et al. 2002). Interestingly, Ceballo et al. (2019) do find an effect of inactivating cortex on discrimination of pure tones and they do not rule out necessity of cortex for finer (<0.585 octaves difference) discriminations. In the present study, we set the discrimination to a “fine” difference of 0.4 octaves. The evidence we present in this paper is consistent with cortex modulating frequency discrimination during associative conditioning but not influencing discrimination – a hypothesis based on a number of prior observations: (1) activating PV interneurons during DFC decreased discrimination following DFC (Aizenberg et al. 2015); (2) cortical inactivation did not affect discrimination when applied after DFC (Aizenberg and Geffen 2013). Here, we found that cortical activity before conditioning predicted the subsequent discrimination ability of the mouse following conditioning but that activity in cortex following conditioning no longer correlated or predicted discrimination ability (Fig 2f & 3d). The model (Fig 7) provides for a mechanism for the relation between auditory cortical activity and associative learning that is consistent both with our results and with prior literature. We discuss these issues in the discussion, lines 613-635.

2) The presentation of the ABR data (Figure S12) is troublesome and actually raises more issues than it solves. In the methods it is stated that ABRs were performed to confirm that the mice had good hearing. "Good hearing" is normally quantified by utilizing an ABR to determine auditory thresholds and these are not presented here. The methods state that broadband clicks at 70 dB SPL were used. To be explicit, you could noise expose a mouse, damaging the cocha, eliciting a permanent shift in auditory threshold, and you would still see an ABR at 70 dB SPL. Furthermore, it is not clear what "maximum ABR amplitude" is meant to indicate - these analyses are typically carried out on individual waves. It would be beneficial to relate specific ABR measurements to learning specificity.

We revisited our ABR data (N = 27 mice) and calculated the ABR threshold across frequency (4-32 kHz) for each mouse (see a in figure below). We found evidence of a weak correlation between threshold (mean threshold of frequencies nearest to CS+ and CS-) and age of ABR measurement (b, Spearman's rank correlation, $r(24) = 0.326$, 95% CI [0.08 0.67], $p = 0.109$), as expected. If the hearing ability of the mice affected their ability to discriminate the CS+ and CS- then one would expect a negative correlation between threshold and learning specificity. For conditioned learner-mice, we did not find a negative correlation between threshold and the learning specificity on retrieval day 1 (Spearman's correlation $r(13) = 0.173$, 95% CI [-0.172 0.81], $p = 0.507$). Furthermore, we found no correlation between the age of the final imaging session and learning specificity (c, including 'non-learners', Spearman's correlation $r(17) = -0.130$, 95% CI [-0.45 0.41], $p = 0.605$). While we endeavored to record the ABRs soon after imaging stopped, some ABRs were not measured until >2 weeks after the final imaging session and thus it is likely that the ABR thresholds are somewhat of an overestimate for these mice (N = 8). Mice whose threshold was >70dB were excluded. All remaining mice included in the study had ABR

thresholds at or below the level of stimulus presentation (70 dB). We have updated figure S12 (now Figure S16) as below and edited the text in the methods.

(Lines 672-675) “The Auditory Brainstem Response (ABR) to tone pips (4 – 32 kHz, 10 – 80 dB SPL) was acquired before or at the end of the experiment, when possible, in order to confirm that mice had thresholds for the stimuli at or below the presentation level (Fig S16). Mice with ABRs > 70 dB were excluded from the study (N=2).”

Figure S16: Auditory Brainstem Responses. (a) Example ABR responses to 5 frequencies presented at 7 different levels. (b) Relationship between age at ABR threshold measurement and threshold (mean of frequencies closest to CS+ and CS-). Mice with threshold greater than 70dB (red) were excluded from the study. (c) Relationship between age at the last imaging session and learning specificity. Black lines in b and c are best linear fits. Statistics: Spearman’s rank correlation.

Minor comments

1) Might there be any non-biological reasons to observe a drop in responsivity? (Surgical quality and whatnot...).

We only observe a drop in responsivity in pseudo-conditioned mice. This is consistent with habituation observed in (Gillet et al. 2018)

2) Lines 168 - 178 seems like an important finding, is there an associated figure?

We added reference to figure 2f to this paragraph. (Lines 174-193)

3) Can any of the changes be due to differences between GCaMP6s and m? Which mice were injected with which?

We performed this analysis (Figure R3). We used GCaMP6s in 7 of the conditioned mice, one of which was subsequently excluded because it did not discriminate following DFC (i.e. was a non-learner). We found that there was no difference in the mean significant Zdiff between mice with the two different GCaMPs (unpaired t-

test, $t_{(21)} = 1.04$, $p = .309$). We have added this information to the methods.

(Lines 802-806) “We used average Z_{diff} across pre-DFC sessions of learner mice to test whether there was a difference between using GCaMP6s (6/23) and GCaMP6m (17/23), we found no difference (unpaired t -test, $t_{(21)} = 1.04$, $p = .309$) between the mean Z_{diff} scores of the two groups of mice and thus we have analyzed them together.”

Figure R3. Mean Z_{diff} of neurons across pre-DFC sessions of mice injected with GCaMP6m ($N = 17$) or GCaMP6s ($N = 6$).

4) Are there any age-related differences? The methods state that, in addition to a number of Cre lines, the *Cdh-23* strain was used. If the max age (27.1 weeks) applies to a Cre line with a C57 background, might that be problematic?

We excluded mice with ABR thresholds greater than 70 dB (see response to point 3 of reviewer 1), which included one C57-cre mouse (*PV-Rosa*, a pseudo-conditioned mouse). We did not find a correlation between the age at the last imaging session of conditioned mice and the learning specificity (Spearman's rank correlation Spearman's correlation $r(17) = -0.130$, 95% CI [-0.45 0.41], $p = 0.605$), suggesting that age does not affect the learning specificity. See response to Reviewer 3 point 3 and panel c in Supplementary figure 16.

5) Figure 3 A&B seems to have the legend text switched?

Thank you for pointing this out, we have corrected the legend.

6) Line 503. It seems like a bit of a stretch to say "We imaged neuronal activity in layers 2/3 because those are the output layers of the cortex, and therefore the plastic changes that occur within the cortex should be evident in these layers." Layer 5&6 are typically referred to as the output layers of cortex. L2/3 do contain a subset of IT-type cells, but they do not in any way reflect the bulk of cortical output.

We have edited this paragraph (Lines 604-612): “Our results relied on tracking the neuronal responses in all transfected neurons in AC without distinguishing between different neuronal subtypes. Previous studies found that a specific class of inhibitory neuron increases activity with presentation of repeated tones (Gillet et al. 2018; Kato et al. 2015). It is therefore plausible that our results include a subset of neurons that function differently during fear conditioning but which we are unable to identify due to lack of selective labelling. Furthermore, we restricted our recordings to layers 2 and 3 of the auditory cortex, and it is possible our results overlook more specific changes in the thalamo-recipient layers of the cortex (Linden and Schreiner 2003; Petrus et al. 2015). The complexity of transformations in the cortical microcircuit and between layers with learning can be explored further (Blackwell and Geffen 2017; Krug 2020; Müsch et al. 2020; Wood et al. 2017).”

Reviewer #2

This manuscript presents a study of neural correlates of learning specificity during fear conditioning. Mice were conditioned to associate one out of two tones with a foot shock, and the specificity of the resulting conditioning was quantified by the difference in freezing following the two tones during subsequent retrieval sessions. Neural activity in the auditory cortex was recorded using calcium imaging before and after conditioning, with a large number of neurons tracked across sessions. The discriminability of the two stimuli based on neural responses was then compared with the specificity of learning across animals.

The main result of the paper is that neural discriminability in A1 before conditioning predicted the learning specificity across animals, while neural discriminability after conditioning did not.

The collected dataset is impressive, and the conclusions of the analysis are intriguing. The manuscript is mostly clearly written (with a couple of exceptions outlined below). My main issue is that I have been struggling to fit the various analyses into a coherent mechanistic view. Additional controls might help clarify the overall picture. Unfortunately, the proposed model does not help clarify potential mechanisms underlying the main finding.

Comments:

(1) The main finding is that neural discriminability before differential fear conditioning (DFC) predicts learning specificity across animals in the first retrieval session, but neural discriminability after DFC did not predict learning specificity in retrieval session 4. There are two possible explanations:

- (i) the neural encoding of the two sounds is different before and after DFC; comparisons of neural discriminability across session however suggest this is not the case (Fig S5).
- (ii) the pattern of learning specificity across animals changes between retrieval sessions 1 and 4.

As far as I could see the second explanation was not investigated. It would be important to show how stable is the pattern of learning specificity across animals from one retrieval session to the next, and explicitly contrast the two explanations outlines above.

This is a good point. We conducted additional analyses to investigate the second explanation. Whereas there is variability in the freezing response over the 4 retrieval sessions (as can be seen in Fig. 1D and S3B), statistically the freezing responses do not change over time (Table S1, post-hoc testing on learning specificity at each retrieval session show no significant differences). Furthermore, a Wilcoxon signed-rank test found no difference between the median learning specificity of retrieval sessions 1 and 4 ($Z = 51$, $p = 0.639$). To get at this question further, we tested the correlation between Zdiff and learning specificity in both directions. If the freezing response of animals remains consistent then we would expect that Zdiff from imaging sessions 2-4 (pre-DFC) would predict learning specificity in retrieval session 4 in addition to retrieval session 1. Using the mean Zdiff of all responsive neurons, we found this to be the case, as indicated in figure f below (purple dots). We reasoned that if Zdiff were rearranged but not learning specificity then the Zdiff post-DFC ought not to predict learning specificity from earlier sessions, we found that Zdiff from imaging session 6-8 (post-DFC) did not predict learning specificity from retrieval session 1 (orange dots, Fig 2f)

We found the same pattern of results using the SVM to measure neural discriminability (figure d below, Fig 3d). Across both measures, the correlations between pre-DFC neural discriminability and learning specificity in retrieval session 4 were stronger than post-DFC neural discriminability and learning specificity in retrieval session 1. This is consistent with the hypothesis that neural activity is reorganized following DFC. Additionally, we found that neural discriminability (Zdiff/SVM performance) is maintained in conditioned animals, but no longer predicts or correlates with learning specificity following DFC. This is consistent with the hypothesis that neural activity is reorganized following DFC and that auditory cortex can no longer modulate the freezing response following conditioning, as suggested by previous work showing that learning specificity is not dependent on auditory cortical activity after fear conditioning (Aizenberg & Geffen 2013).

Figure 2f: Spearman's rank correlation ($\rho \pm 95\%$ CI) between Z_{diff} score averaged across 3 imaging sessions as indicated in the legend and learning specificity from retrieval sessions 1 and 4. Dots represent individual bootstrapped ρ ($n = 1000$). $^\dagger p < 0.1$, $*p < 0.05$, $**p < 0.01$, $***p < 0.001$, $^{n.s.} p > 0.10$.

Figure 3d: Spearman's rank correlation ($\rho \pm 95\%$ CI) between SVM performance averaged across 3 imaging sessions preceding retrieval sessions 1 (blue), and 4 (orange). Dots represent individual bootstrapped correlation values ($n = 1000$). $^\dagger p < 0.1$, $*p < 0.05$, $**p < 0.01$, $***p < 0.001$, $^{n.s.} p > 0.10$.

Additionally, I have found the analyses in Fig S5 difficult to read. It would be useful to show for example a confusion matrix where a decoder is trained on one session and tested on another.

We re-worked figure S5 – now figure S6 - to include confusion matrices. The findings show that on average, neuronal discriminability decreased over sessions in pseudo-conditioned mice but not in conditioned mice.

Lines 315-344: “To further investigate how neuronal discrimination changed over time, we tested the neuronal discrimination performance of the SVM using cells tracked across pairs of imaging sessions. We trained the SVM using one imaging session and tested on data held out from that session and from the same cells in the second testing session (Fig S6a & b). If neuronal discriminability is maintained in conditioned mice, we would expect that there would be no change in performance between training and testing sessions. By contrast, in pseudo-conditioned mice, as neuronal discriminability appears to decrease, we expected to observe a decrease in performance particularly between sessions pre- and post-DFC. In conditioned mice, there was a small deficit in the testing sessions compared with training sessions, which did not change over sessions. By contrast, in pseudo-conditioned mice, we observed the same deficit in testing sessions compared with training, but the deficit increased over sessions. A linear regression of difference in performance with mouse group (m) and # sessions between testing and training (s) as predictors indicated that the slope of the relationship was significantly different between conditioned and pseudo-conditioned mice (Table S1, $m*s$, $p = .020$). Similarly, we observed a decrease in Z_{diff} as the number of sessions between pairs increased in pseudo-conditioned mice, but not in conditioned mice (Fig S6c & d, Table S1, Linear regression, $m*s$, $p = .004$). Neuronal representations are stabilized over time with behavioral relevance and drift without^{26,27}. To assess whether representation of the CS+ and CS- was stabilized in conditioned vs. pseudo-conditioned mice we investigated whether was drift in the Z_{diff} of populations of neurons. If there is drift in the neuronal representation, then the similarity of Z_{diff} between individual neurons over time should become progressively dissimilar. We calculated the similarity (Pearson's correlation) of Z_{diff} scores of neurons tracked between pairs of imaging sessions (Fig S6e). We fit a linear mixed-effects model to predict how Z_{diff} similarity between sessions was affected by the time between imaging sessions and whether mice were conditioned or pseudo-conditioned. We found there was a negative effect of number of imaging sessions between pairs of sessions on Z_{diff} similarity (Table S1, $t(624) = -2.87$, $p = .004$) but no difference in the effect between conditioned and pseudo-conditioned mice ($t(624) = 1.28$, $p = .201$). In summary, there is evidence of drift in the Z_{diff} score of both groups of mice, indicating that the Z_{diff} of individual cells became progressively dissimilar. In conditioned mice, the average Z_{diff} was maintained, while in pseudo-conditioned mice it decreased.”

Figure S6: Changes in neuronal discrimination between imaging sessions. (a) The SVM was trained with data of neurons tracked between a pair of sessions from one of the sessions. The SVM was subsequently with data left out from that training set (10-fold cross validation) and with the same number of trials from the left-out neurons using the testing set. This was repeated across all pairs of imaging sessions. These graphs show the difference in performance between the training and the testing sets for each pair of imaging sessions. (b) The relationship between the difference in performance from a and the number of sessions between each pair in the forward direction (upper triangle outlined in a). Black lines show best linear fit. (c) The difference in Zdiff between neurons tracked between pairs of imaging sessions for conditioned and pseudo-conditioned mice. (d) The relationship between difference in Zdiff and number of sessions between each pair in the forward direction (upper triangle outlined in c). (e) Similarity of the Zdiff scores of neurons tracked between pairs of sessions were assessed using Pearson correlation (ρ). The panels show the similarity between each pair of sessions averaged across conditioned (left) and pseudo-conditioned mice (right). Black lines show best linear fit. Statistics: Spearman's rank correlation, $\dagger p < 0.1$, $*p < .05$, $**p < .01$, $***p < 0.001$, $n.s. p > .05$.

2) It is somewhat surprising that learning specificity is better predicted by average Z-scores than by population discriminability. The explanation provided on 1.213-215 does not real hold, as the SVM should give vanishing weights to neurons with non-specific responses. Conversely, the Z-score was averaged only across significant neurons. How is this statistically justified? Would the results hold if the Z-score was averaged across all neurons?

Yes, thank you for noticing this. The SVM should give smaller weights to non-important neurons. We have altered the figures so that in figure 2 we are now presenting Zdiff of all neurons whether they are significant or not. This does not affect the main findings. Additionally, we compared the correlations between SVM performance and Zdiff with the same SVM performance and mean significant Zdiff scores, finding no difference in the correlations. We have added the following lines to the text and deleted the lines mentioned above:

Lines 259-262: "Since the SVM should give greater weight to more informative neurons, we tested whether there would be a stronger correlation between the significant Zdiff scores and SVM performance. We found that the correlations were not significantly different (bootstrap comparison, ρ difference = -0.044, 95% CI [-0.28, 0.14], $p = .562$)."

3) The model does not address the main finding of the paper, i.e. the fact that neural discriminability predicts learning specificity before but not after DFC. The model shows that higher neural discriminability leads to higher

learning specificity, which seems unsurprising. The model is then used to address two findings from other papers that do not seem directly related to the main result. Unless the model is used to test a mechanism reconciling the different findings (see point 1 above), it should be left out.

The model was included to reconcile the present results with prior findings. We expanded the model to show that one explanation for the change in correlation between discriminability and learning specificity following conditioning can be due to small drift in neuronal response properties as showed previously (Clopath et al. 2017).

*Lines 865-871: "In the simulations, we tested for two different values of initial σ^{ctx} ; one corresponding to narrow tuning with a small overlap ($\sigma^{ctx} = 3$), and one corresponding to broad tuning with a large overlap ($\sigma^{ctx} = 10$). (Note that $\sigma^{MGB} = 0.8$ was equivalent to $\sigma^{ctx} = 3$ since we did not model MGB inhibition here, $I^{MGB} = 0$). To avoid boundary effects, we had a circular boundary condition of the 10 inputs, meaning that input 1 and input 10 are neighbors. We also assumed that the tuning σ^{ctx} would drift over time. **Specifically, at every time step, we added a uniform random noise between -0.25 and 0.25 to σ^{ctx} .** σ^{ctx} was bounded between 4 and 20."*

4) There seems to be a mismatch between the panels D and E in Fig 2, and the corresponding legend. The legends are likely swapped, but more importantly in D "does not correlate" should presumably be "does correlate".

Thank you for spotting this error, we have corrected this.

5) Similar issue with panels A and B in Fig 3. \pm

Thank you, we have corrected this.

6) The paragraph on l.278-286 seems to be a repetition of the result presented in the paragraph on l. 168-178. I was confused by the logic here.

The paragraph in lines 168-178 (now lines 174-193) refers to whether neuronal discrimination before conditioning is able to predict the subsequent learning specificity. We find that it does (Fig 2d). To check whether, following conditioning, it still does, we test whether there is a prediction of learning specificity in retrieval session 4 from neuronal discriminability before retrieval session 4 but after conditioning. By contrast, lines 278-286 (now lines 346-353) refer to whether the neuronal discriminability predicts learning specificity recorded on the same days: imaging sessions 5-8 and learning specificity from retrieval sessions 1-4, which occur on the same days respectively.

7) l.88-90: there is some issue with brackets.

Thank you, we have edited this sentence.

Reviewer #3 (Remarks to the Author):

In this manuscript, Wood et al. explore how auditory cortical activity is correlated with differential fear learning in mice. Specifically, the authors combine a tone based Differential Fear Conditioning (DFC) protocol with longitudinal two-photon imaging in the auditory cortex of mice to ask how average neural responses prior to and after DFC correlate with individual learning specificity. They observe that average neuronal discriminability between CS+ and CS- tone frequencies prior to DFC is significantly correlated with fear learning specificity. However, such a correlation is not observed post DFC suggesting that auditory cortical activity post DFC does not encode differential fear memory. Furthermore, while there are changes in the normalized neural tunings following DFC, these changes do not correlate with fear learning specificity. The authors finally use a computational model of the auditory cortex, MGB and the basolateral amygdala to test their findings and reconcile them with previous findings in this area that reinforce the role of the auditory cortex in fear learning specificity prior to but not post DFC.

While this paper explores an interesting issue, there are some key concerns that need to be addressed before these results can be fully appreciated. Additionally, the manuscript as it stands now does not take full advantage of the longitudinal imaging strategy that can be used to address deeper questions about single neuron changes pre and post DFC. Here are my concerns below:

Major concerns:

1) The authors need to make sure that their results are not biased by neuronal sampling variability across different animals. It is important to provide basic sampling information and make sure they are comparable across mice - the number of neurons sampled per mouse, number of neurons with significant z_{diff} per mouse, variability in best frequency distributions of sampled neurons between mice. Additionally, the authors need to ensure that their results are not biased by the region of imaging. One way to do this would be to image from multiple fields in the same mouse and compare neuron response properties across different fields, to verify if results are robust to variability in the region of sampling.

This is a good point. We have conducted additional analyses to address the question of whether the variability in number of neurons sampled between mice affected the results. We have implemented a resampling method for calculating Z_{diff} and SVM performance. We calculated the minimum number of responsive neurons across mice and used this value to resample with replacement neurons from each mouse (x100). This removes the sampling bias by using the same number of neurons from each mouse. The implementation of this method did not affect the main findings of the paper (Figures 2-4).

Lines 153-157: "To test whether neuronal discrimination pre-DFC could predict subsequent learning specificity, we averaged the Z_{diff} scores of neurons recorded in each recording session of each mouse and compared it with learning specificity 24 hours post-DFC. Since different numbers of neurons were recorded from each mouse, we resampled (100x with replacement) the lowest number of neurons recorded from across the mice."

Lines 243-246: "We trained a Support Vector Machine (SVM) to discriminate between presentation of CS+ and CS- using population responses to the two stimuli – again we resampled (100x with replacement) the lowest number of neurons recorded from across the mice."

To address whether the variability in best frequency distributions between mice affected the results, we have implemented a similar resampling method for calculating the change in response of tracked neurons (Figures 5-6). We calculated the minimum number of neurons with best frequencies at each of the presented frequencies across the conditioned and pseudo-conditioned mice. We then used this value to resample with replacement neurons with best frequencies in each frequency bin for conditioned and pseudo-conditioned mice. This has the effect of normalizing the pre-DFC best frequency distribution between conditioned and pseudo-conditioned mice (Figure S10). We found that there was still an increase in normalized response at CS+ in conditioned mice and no change in response at either CS in pseudo-conditioned mice. There was an increase in distance of best

frequency from CS+ in pseudo-conditioned mice but not in conditioned mice, which was different to the main finding in figure 6.

Lines 434-451: “To verify that the results were robust to variability in frequency tuning between conditioned and pseudo-conditioned mice, we performed the analysis on change in response, change in distance of best frequency from CS+, and change in sparseness resampling the same number of neurons from each best frequency bin (1/12 bins). This had the effect of normalizing the frequency distributions pre-DFC between conditioned and pseudo-conditioned mice. We found that there was still an increase in response at the CS+ in conditioned mice while there were no changes at CS-, and no changes at either CS in pseudo-conditioned mice (Fig S10a). Furthermore, we found that despite the increase in response at CS+ in conditioned mice, there was no change in the distance of best frequency from the CS+ on average, whilst there was an increase in distance from CS+ in pseudo-conditioned mice (Fig S10b) while sparseness fell in both groups of mice (Fig S10c). Observing the best frequency distributions post-DFC (Fig S10d), this change is driven mostly an increase in neurons with best frequency at the extremes of our measurement (5 and 32 kHz). Qualitatively, the conditioned mice showed increased numbers of neurons tuned at and above the CS+ and decreased numbers below CS+ compared with pseudo-conditioned mice. Pairing of the CS+ with the shock led to increased number of neurons tuned to frequencies at and above the CS+ compared with an unpaired shock. Combined, whereas we find some changes in tuning consistent with classical results, these changes do not account for the differential learning specificity across mice.”

With regards to imaging region, we have examined the main metrics with regards to location on the anterior-posterior axis: see response to Reviewer 1, point 1. Using the same resampling methodologies as above, we examined the main findings of the paper with regard as to whether the conditioned mice had imaging windows located in putative A1 or AAF (Figure S10). We found that there was no difference between the two putative regions with regards to predicting learning specificity, both Zdiff and SVM performance predicted learning specificity of mice in each region. And there was no change in Zdiff from pre- to post-DFC. We did, however, find a difference in the change in response to CS+, with neurons in A1 exhibiting a change in response at CS+ while neurons in AAF did not. However, this change in response at CS+ in A1 did not correlate with learning specificity.

Lines 452-464: “To investigate whether variability in the region of sampling in each mouse affected the main findings, we split the mice into two groups based on whether the location that the center of their imaging window mapped onto the anterior-posterior axis. Locations that also contained the auditory thalamus (medial geniculate body, Fig S1) or not, with each group’s field of view more likely to be from primary auditory cortex (A1) or the anterior auditory field (AAF), respectively(Hackett et al. 2011). We found that the changes in response at CS+ were driven by neurons in putative A1 where there was a significant increase in normalized response and not in putative AAF where there was no change in response (Fig S11a). The distance of best frequency increased on average in AAF while there was no change in A1 (Fig S11b). However, we found no effect of imaging region on prediction of learning specificity by Zdiff or the SVM performance pre-DFC (Fig S11c-d) and no effect of imaging region on change in Zdiff (Fig S11e). Thus, there appears to be a differential effect of change in response at CS+ following conditioning for primary regions A1 and AAF, but this does not appear strongly related to the learning specificity (Fig S11f).”

We found no correlation between the A-P location relative to bregma and the number of responsive neurons in the imaging window pre-DFC or with the % of neurons with significant Zdiff pre-DFC. There was furthermore no correlation between mean best frequency and mean Zdiff pre-DFC. Together, these results suggest that the results of the study were not influenced by the region of imaging or the position of the field of view along the anterior-posterior axis. We have added a new supplementary figure (S5) and the following text:

Lines 194-215: “To verify that the results were robust to variability in frequency tuning distributions and location of the imaging window along the anterior-posterior axis (Fig S1 & S2) between mice, we investigated the relationship between Zdiff and these parameters. If neuronal discriminability is affected by imaging location then we would expect a relationship between the location of the imaging field of view on the anterior-posterior axis and Zdiff, we did not find any relationship between these two measures (Fig S5a, Spearman’s rank correlation, $p > .05$) nor between the percentage of neurons with significant Zdiff and imaging location (Fig S5b). The best frequency distributions of neurons in the imaging window could affect the mean Zdiff of neurons of each mouse,

if so we would expect higher Z_{diff} scores and more neurons with significant Z_{diff} scores for neurons tuned around the CS+ and CS-. However, we found no relationship between mean Z_{diff} score and mean best frequency in the imaging window (Fig S5c, Spearman's rank correlation, $p > .05$) nor between the percentage of significant Z_{diff} scores and mean best frequency (Fig S5d). We did find a relationship between the percentage of significant Z_{diff} scores and learning specificity (Spearman's rank correlation, $\rho = .72$, $p = .011$) suggesting that the best discriminating mice also had more neurons that discriminated between CS+ and CS- (Fig S5e). While there was no relationship between mean Z_{diff} and mean best frequency of all neurons in the imaging window across mice, we did find that neurons with best frequency at CS+ or CS- had higher Z_{diff} scores than neurons tuned to other frequencies (Fig S5f, Table S1). This suggests that mice with more neurons with best frequencies at CS+ and CS- might have better learning specificity however, we found no relationship between the percentage of neurons in the imaging window with best frequency at CS+ and CS- across the pre-DFC imaging sessions and learning specificity (Spearman's rank correlation, $\rho(12) = .46$, 95% CI [-.06, .71], $p = .127$.)”

Figure S5: Relationship between Z_{diff} and best frequency distributions and anatomical location. (a) Number of responsive neurons with significant Z_{diff} and Z_{diff} of responsive neurons was averaged in each session and across sessions for each mouse. Relationship between the distance from bregma of the imaging field of view and the mean Z_{diff} scores of conditioned (black circles) and pseudo-conditioned mice (red diamonds) across pre-DFC sessions. (c) Relationship between the distance from bregma of the imaging field of view and the % of significant Z_{diff} scores across pre-DFC sessions. (d) Relationship between mean best frequency of responsive units in the imaging field of view and the mean Z_{diff} scores across pre-DFC sessions. (e) Relationship between mean best frequency of responsive units in the imaging field of view and the % of significant Z_{diff} scores across pre-DFC sessions. (f) Relationship between the mean % of significant Z_{diff} scores in the imaging field of view across pre-DFC sessions and the learning specificity from retrieval session 1 for conditioned mice. (g) Relationship between mean Z_{diff} across tracked pre-DFC sessions for each neuron and best frequency. Grey bars show median $Z_{diff} \pm sem$. Statistics: Spearman's rank correlation.

2) It is interesting that average z_{diff} does not change between consecutive imaging sessions, and between pre and post DFC. The authors also show that learning specificity is stable across retrieval trials post DFC. Given this general stability both at the neural and behavioral level, it is a little confusing to me that while pre Z_{diff} is correlated with learning specificity, post Z_{diff} is not. How do the authors explain this?

The Z_{diff} does not change on average pre- to post-DFC in conditioned mice. In pseudo-conditioned mice, there is a decrease in Z_{diff} from pre- to post-DFC (Fig 4, S5). The average Z_{diff} is maintained in conditioned mice but

does not correlate with learning specificity, also stable, after DFC. We have expanded the model to include a possible explanation as a small random drift in the response properties neurons that support neuronal discrimination (Clopath et al. 2017) (Fig.7). While the population average of Z_{diff} is maintained over time, individual neurons carrying that Z_{diff} change. This is supported by changes in the numbers of neurons that have significant Z_{diff} from pre- to post-DFC (Figure R4). Whereas a small proportion of neurons maintains significant Z_{diff} from pre- to post-DFC, there is a significant proportion of neurons in conditioned and pseudo-conditioned mice that switch from significant to not and vice versa.

Figure R4. Percentage of neurons with significant Z_{diff} in conditioned (left) and pseudo-conditioned (right) mice for pre-DFC (dark blue), post-DFC (cyan), both pre-DFC and post-DFC (green) or nor significant (yellow).

3) Given that the authors have measured and tracked single neuron responses longitudinally, it would be useful to see some measures of response stability/variability for individual neurons across days (both during baseline and post DFC).

(i) How stable are z_{diff} measures for individual neurons across baseline days and do individual neurons maintain their z_{diff} following DFC or are new neurons recruited?

We performed this analysis. Z_{diff} measures are relatively stable across days pre-DFC as indicated by the low standard deviation observed across days of neurons tracked over 4 sessions pre-DFC (Figure R5). The Z_{diff} variability between days of the same neurons is generally lower than the variability between neurons on a given day. Furthermore, variability did not change after conditioning, indicated by the geometric coefficient of variance being the same pre- and post-DFC among neurons tracked for all sessions pre- or post-DFC.

To investigate whether the same or different neurons maintain the Z_{diff} from pre- to post-DFC, we tested the % of neurons with significant Z_{diff} pre- and post-DFC and whether they were significant in one or both times (Figure R4). We found that about half of the neurons with significant Z_{diff} pre-DFC for conditioned ($N = 818$) and pseudo-conditioned ($N = 580$) mice did not have significant Z_{diff} post-DFC. About half of those that were significant pre-DFC maintained that post-DFC. In summary, while Z_{diff} is generally consistent across days and a core of neurons maintains significant discriminability from pre- to post-DFC, there is a large proportion of neurons that lose or gain significant discriminability.

Figure R5. Variability and stability of Zdiff across days. Distribution of Zdiff for neurons across days (variability, left; mean, center). Zdiff variability and mean across neurons is indicated by the red lines. Right: variance of neurons pre- and post- DFC does not change.

(ii) Similarly, what is the stability of frequency tuning across days for a single neuron at baseline?

We performed this analysis. Frequency tuning was also relatively stable across neurons pre-DFC with the majority of neurons having a σ_{BF} of less than the standard deviation between neurons (Figure R6). There was however a strong relationship between the tuning sharpness or strength, as measured by the sparseness, and the σ_{BF} of each neuron (Figure R6 right). The more sharply tuned a neuron was the lower the σ_{BF} . Thus, sharply tuned neurons tended to have more stable best frequencies.

Figure R6. Variability in frequency tuning across days. Left: Distribution of variability in best frequency across days. Right: Variability in best frequency versus the mean sparseness across days.

(iii) Do individual neurons increase their responses to CS+ tone frequency following DFC or are new neurons that were previously not responsive to CS+ recruited?

We performed this analysis. The number of neurons significantly responsive to CS+ decreases following DFC, with 16% of neurons significantly responsive pre-DFC and 9% responsive post-DFC (Figure R7). 16% of neurons were significantly responsive to CS+ pre- and post-DFC while 59% of neurons were not significantly responsive to CS+ pre- or post-DFC. As can be seen in supplementary figure 6, the average response to CS+ does not change for conditioned mice. However, when the tuning curves of neurons are normalized to between 0 and 1, we can see that there is an increase in response at CS+ (Fig. 5), indicating that the relative response at CS+ compared with other frequencies has increased in conditioned mice.

Figure R7. Percentage of neurons significantly responsive to CS+: only pre-DFC (blue); only post-DFC (cyan); both pre- and post-DFC (green); not significant.

4) In figure 5, while the authors have taken population averages of neuronal tuning profiles across mice, it is important to take into consideration other changes that could be influencing population responses. For instance, do the proportions of neurons responding to CS+ or CS- change post DFC, and is this correlated with learning specificity?

We performed this analysis (Figure R8). Neurons with responses different from zero (t-test, $p < 0.05$) at CS+ or CS- were defined as having significant responses. We found that on average there was no significant change in the proportion of significantly responsive neurons, tracked over days, from pre- to post-DFC. Furthermore, the change in proportion of responsive neurons to CS+ or CS- did not correlate with the learning specificity (Spearman's rank correlation, $p > 0.05$).

Figure R8. Left: Proportion of significant responses to CS- or CS+ pre (blue) and post (red) DFC. Right: Learning specificity does not correlate with change in proportion of neurons that are significantly responsive to either CS+ (magenta) or CS- (cyan).

5) Similarly, what are the proportions of neurons with significant zdiffs prior to and post DFC, and are they correlated to learning specificity in figures 2 and 4.

Average number of neurons that respond significantly in each imaging window across pre-DFC imaging sessions does not correlate with the learning specificity (Spearman's rank correlation, $r = 0.108, p = 0.669$). However, the average proportion of responsive neurons with significant Zdiff across pre-DFC imaging sessions does correlate with learning specificity (See response to Reviewer 3 point 1 and new supplementary figure S5). As expected, the more neurons a mouse has that discriminate between the CS+ and CS- pre-DFC, the better the mouse subsequently discriminates the CS+ and CS- following differential fear conditioning.

6) In Figure 6, do changes in best tone frequency distance from CS+ for a given mouse correlate with its learning specificity? Do the changes observed depend on the baseline distribution of best frequencies for a mouse? We have addressed this question by performing a new analysis as described in response to Reviewer 3 point 1, we effectively normalize the pre-DFC best frequency distribution to a flat distribution.

7) The authors discuss their rationale for choosing tone stimuli in their DFC experiments as opposed to more complex auditory stimuli. The authors can more directly address the debate concerning the role of the auditory

cortex in differential fear learning of simple vs complex auditory stimuli by testing one additional question in their final model – does suppressing the auditory cortex during DFC impact learning specificity?

We added an extra condition to our model where we set the auditory cortex current to zero during the conditioning phase (see methods) as an analogue of cortical inactivation during conditioning. We found that in the modelled learning specificity after conditioning there was still discrimination however the specificity of learning was reduced compared to the full model. These results mirror those found by Dalmy et al. (2019) who tested freezing after performing DFC with optogenetic cortical activation. They found that optogenetically manipulated animals showed reduced freezing to CS+ and increased freezing to CS-, and therefore reduced discriminability/learning specificity. The model replicated the change in learning specificity.

Figure 7c: Normalized learning specificity at two AC discriminability levels; fine (light blue) and broad (dark blue) tuning. Results are shown for learning specificity with no interventions, when inhibition is reduced in AC during DFC (analogue of when ArchT-transfected PV interneurons in AC are inactivated by optogenetics during DFC), when AC is inactivated during DFC (in the model), and when AC is inactivated during memory recall (analogue of an injection of muscimol during memory recall; PV = parvalbumin positive interneurons, ArchT = Archaelhodopsin-T).

1) Adding to the current supplementary data in the paper, are mean response magnitudes to CS+, CS-, or their difference (normalized or absolute) post DFC correlated with learning specificity?

We performed the additional analyses comparing normalized responses to CS+ and CS- with learning specificity and found no relationship. We have added the panels below to Figure S8.

Figure S8. (e) Normalized response magnitude to CS+ post-DFC against mean learning specificity across retrieval sessions 1-4. Magenta line represents the best linear fit. (f) Same as e but for response to CS-. (g) Same as e but for difference in normalized response magnitude between CS+ and CS-. Statistics: Spearman's correlation ($N = 14$). Data are shown as mean \pm sem.

2) How do the authors calculate average zdiff values over multiple days? Do they only include neurons that can be successfully tracked and that have significant zdiff responses across all sessions or do these come from different populations of neurons?

When comparing neurons responsive and tracked between pre- to post-DFC, the Zdiff is calculated within each session and is averaged together for each neuron for all responsive pre- and post-DFC sessions respectively. When comparing averages across mice, Zdiff of responsive neurons was averaged within each session and then across sessions.

3) Legends for Figure 2 D and E seem to be switched. Same for Figure 3A and B.

Thank you for spotting this, we have fixed this issue.

REVIEWERS' COMMENTS

Reviewer #1 (Remarks to the Author):

I want to commend the authors for an absolutely stellar revision. They have clearly went above and beyond to respond to all reviewer comments and this is readily apparent in the response letter. I heartily recommend acceptance.

Reviewer #2 (Remarks to the Author):

I thank the authors for the new analyses provided in the response and revised manuscript. Unfortunately, these analyses do not alleviate my concerns. In particular, the confusion matrices (new Fig S6) do not seem to show any sign of a clear change in neural representation following conditioning, one of the results announced in the abstract ("Stimulus representation in auditory cortex was reorganized following conditioning. "). So I am still struggling to fit the different analyses in a coherent view, and find the proposed conclusions unsupported.

PS: It is actually not clear to me what is plotted in the confusion matrices. I suggested to plot the performance when a decoder is trained on one session and tested on another, a standard way to assess a change in representation. Fig S6 shows the "difference" between trained and tested performance, which is an unusual quantity. I also don't understand how a difference between two decoding performances can take such large negative values and what it means.

Reviewer #3 (Remarks to the Author):

The authors have sufficiently addressed my comments and provided the relevant data to interpret the results of the study.

REVIEWERS' COMMENTS

Reviewer #1 (Remarks to the Author):

I want to commend the authors for an absolutely stellar revision. They have clearly went above and beyond to respond to all reviewer comments and this is readily apparent in the response letter. I heartily recommend acceptance.

Thank you so much for such a lovely response, it is very much appreciated.

Reviewer #2 (Remarks to the Author):

I thank the authors for the new analyses provided in the response and revised manuscript. Unfortunately, these analyses do not alleviate my concerns. In particular, the confusion matrices (new Fig S6) do not seem to show any sign of a clear change in neural representation following conditioning, one of the results announced in the abstract ("Stimulus representation in auditory cortex was reorganized following conditioning. "). So I am still struggling to fit the different analyses in a coherent view, and find the proposed conclusions unsupported.

The changes in stimulus representation in auditory cortex are shown in figure 5, S7 and S10. We found that these changes in stimulus representation did not relate to learning specificity of mice (figure S8).

PS: It is actually not clear to me what is plotted in the confusion matrices. I suggested to plot the performance when a decoder is trained on one session and tested on another, a standard way to assess a change in representation. Fig S6 shows the "difference" between trained and tested performance, which is an unusual quantity. I also don't understand how a difference between two decoding performances can take such large negative values and what it means.

In figure S6a we train and test the SVM decoder using data from cells tracked between a training session and a testing session and then average results across mice. Since the number of neurons tracked across each pair of sessions differs, we show the performance of the SVM decoder on the testing session relative to baseline (performance of the decoder on held out data from the training session).

Figure S6b plots the data outlined in black in Figure S6a, which is the 'forward-in-time' pairs of testing/training as a function of the number of sessions between the training and the testing. We find that for pseudo-conditioned mice there is a decrease in performance as the number of sessions between training and testing sessions increases. In contrast, for conditioned mice, there is no change in performance as a function of the number of sessions. These findings are consistent with data in Figure 4b.

Thus, even though stimulus representation is reorganised following fear conditioning (Figure 5), there is no change in ability to decode the stimulus from the neural data from pre- to post-fear conditioning (Figure 4 & S6).

To make the results in this figure clearer, we have edited the legend for figure S6a and the main text as follows:

Legend: "The SVM was trained (with 10-fold cross validation) with data from a single imaging session (training session) and subsequently tested on data from the same neurons recorded from a second imaging session (testing session). Since the number of neurons tracked between each pair of sessions varied, performance of the SVM on the testing data is shown relative to the baseline of performance of the SVM on data left out from the training session. SVM performance is shown averaged across mice."

Main text, lines 322-326: "To further investigate how neuronal discrimination changed over time, we tested the neuronal discrimination performance of the SVM using cells tracked across pairs of imaging sessions. We trained the SVM using one imaging session and obtained a baseline SVM performance on data held out from the training session. For testing, SVM performance was measured using data from the same neurons in the testing session (Fig S6a & b)."

Reviewer #3 (Remarks to the Author):

The authors have sufficiently addressed my comments and provided the relevant data to interpret the results of the study.

Thank you for your feedback.